# DeepRetrieval: Hacking Real Search Engines and Retrievers with Large Language Models via Reinforcement Learning

**Pengcheng Jiang**[*], **Jiacheng Lin**[*], **Lang Cao, Runchu Tian, SeongKu Kang**[†],
**Zifeng Wang, Jimeng Sun, and Jiawei Han**

University of Illinois Urbana-Champaign     [†]Korea University

## Abstract

Information retrieval systems are crucial for enabling effective access to large document collections. Recent approaches have leveraged Large Language Models (LLMs) to enhance retrieval performance through query augmentation, but often rely on expensive supervised learning or distillation techniques that require significant computational resources and hand-labeled data. We introduce DeepRetrieval, a reinforcement learning (RL) approach that trains LLMs for query generation[1] through trial and error without supervised data (reference query). Using retrieval metrics as rewards, our system generates queries that maximize retrieval performance. Deep-Retrieval outperforms leading methods on literature search with 65.07% (vs. previous SOTA 24.68%) recall for publication search and 63.18% (vs. previous SOTA 32.11%) recall for trial search using real-world search engines. DeepRetrieval also dominates in evidence-seeking retrieval, classic information retrieval and SQL database search. With only 3B parameters, it outperforms industry-leading models like GPT-4o and Claude-3.5-Sonnet on those tasks. These results demonstrate that our RL approach offers a more efficient and effective paradigm for information retrieval. Our data and code are available at: https://github.com/pat-jj/DeepRetrieval.

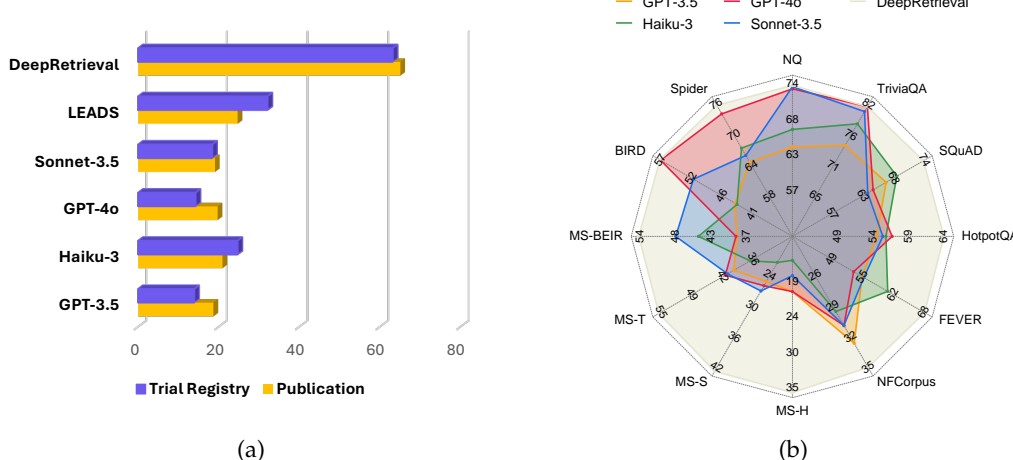

(a)                                    (b)

Figure 1: **Performance Overview of DeepRetrieval. (a)** Literature Search Performance (Recall@3K) on Real-World Search Engines (PubMed and ClinicalTrials.gov); **(b)** Performance on Classic Information Retrieval and SQL database search Tasks.

## 1 Introduction

Information retrieval (IR) systems play a critical role in helping users find relevant information within large document collections. In today's data-rich environment, the effectiveness

---

[*]Major contributors. Contribution statements are provided in Appendix I.

[1]In this study, we use the term "query generation" to cover both "query augmentation/rewriting" for text retrieval and "SQL query generation" for database search.

of these systems directly impacts our ability to access and leverage knowledge across domains ranging from scientific literature and clinical evidence to general web content and structured databases (Baeza-Yates et al., 1999; Singhal et al., 2001).

Traditional approaches to IR rely heavily on keyword matching and statistical methods, which often struggle with understanding the semantic meaning behind user queries (Krovetz & Croft, 1992; Schütze et al., 2008; Robertson et al., 2009). The semantic gap between user information needs and their query formulations has long been recognized as a fundamental challenge in IR (Furnas et al., 1987; Croft et al., 2010). A user's initial query is frequently an imperfect representation of their information need, limited by vocabulary mismatches, domain knowledge gaps, or an inability to articulate complex requirements (Belkin et al., 1982; Ingwersen, 1996; Taylor, 1968).

Recent advances in Large Language Models (LLMs) have shown promising results in addressing these limitations through query augmentation (Bonifacio et al., 2022; Mao et al., 2020; Nogueira & Cho, 2019), where LLMs expand or reformulate user queries to better capture relevant documents. By bridging the semantic gap between users' information needs and document collections, query augmentation helps retrieve more relevant content that might otherwise be missed using traditional keyword matching (Dai & Callan, 2019; Yates et al., 2021; Gao et al., 2021). However, current LLM-based approaches to query augmentation typically employ supervised learning or distillation techniques, which face several significant limitations. They require expensive computational resources to generate training data, often costing thousands of dollars for instruction tuning or distillation from larger models (Wang et al., 2022c; Zhou et al., 2023; Wang et al., 2025). The quality of augmented queries depends heavily on the quality and coverage of supervision data (Khattab et al., 2021), and they rely on larger models to generate data for smaller ones, potentially introducing biases and limitations (Beyer et al., 2022). Furthermore, these approaches typically optimize for mimicking human-generated or teacher model-generated queries rather than for end-task performance (Kudugunta et al., 2021; Asai et al., 2023).

In this work, we introduce DeepRetrieval, a novel approach that uses reinforcement learning (RL) to train LLMs for query generation/rewriting. Rather than relying on supervision data, DeepRetrieval allows the model to learn through direct trial and error, using retrieval metrics (such as NDCG (Järvelin & Kekäläinen, 2002)) as reward. Our approach is inspired by recent advances in using RL to enhance reasoning capabilities in LLMs (Guo et al., 2025; Lightman et al., 2023; Hsieh et al., 2023), but specifically adapted for the IR domain.

We evaluate DeepRetrieval across a diverse range of retrieval tasks, including literature search using **real-world search engines** (PubMed and ClinicalTrials.gov), evidence-seeking retrieval across common question-answering datasets (Kwiatkowski et al., 2019; Joshi et al., 2017; Rajpurkar et al., 2016), classic information retrieval benchmarks with BM25 (Robertson et al., 2009) and dense retrieval systems (Karpukhin et al., 2020), and SQL database search for structured data access (Yu et al., 2018). DeepRetrieval significantly outperforms state-of-the-art models on literature search. It achieves 65.07% recall (vs. previous SOTA of 24.68%) for publication search and 63.18% (vs. previous SOTA of 32.11%) for clinical trial search (Wang et al., 2025). Also, it demonstrates superior performance on evidence-seeking retrieval (Ma et al., 2021; Yates et al., 2021), classic IR tasks (Thakur et al., 2021), and SQL database search (Gao et al., 2023a), outperforming even industry-leading models like GPT-4o (Hurst et al., 2024) and Claude-3.5-Sonnet (Anthropic, 2024) with only 3B parameters.

The key contributions of our work include:

[1] A novel reinforcement learning framework for query generation that directly optimizes for retrieval performance, which is a more cost-efficient paradigm for information retrieval that eliminates the need for expensive supervision data for query generation

[2] A structured generation method that enables effective reasoning before query formulation. We conduct comprehensive evaluation across diverse retrieval tasks demonstrating consistent superiority over existing methods

[3] We demonstrate that RL offers a more efficient and effective paradigm for enhancing IR systems, potentially changing how we approach the fundamental challenge of connecting users with relevant information across different domains and retrieval settings

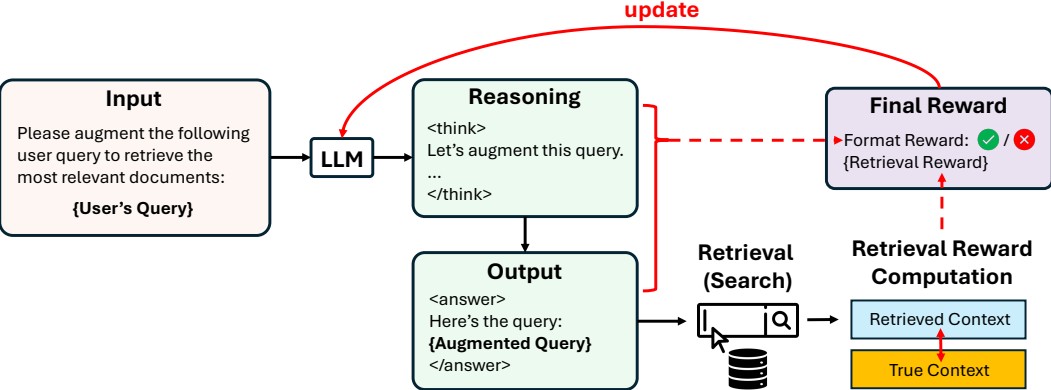

Figure 2: **Abstractive Overview of DeepRetrieval**: Based on an input user query, the LLM generates an augmented query, which is used to retrieve documents. Format reward and retrieval reward are both computed as the feedback to update the model.

## 2 DeepRetrieval

DeepRetrieval is an RL-based framework for training LLMs to enhance information retrieval through query generation/rewriting. Unlike existing approaches (Lin et al., 2020; Ye et al., 2023; Ma et al., 2023; Wang et al., 2024b) that rely on supervised learning with labeled query-augmentation pairs, our approach learns directly from retrieval outcomes. Figure 2 illustrates the overall architecture of our system.

### 2.1 Problem Formulation

Let $D$ be a collection of contexts and $q$ be a user query. The goal of an information retrieval system is to return a set of contexts $D_q \subset D$ that are relevant to $q$. In query augmentation, the original query $q$ is transformed into an augmented query $q'$ that is more effective in retrieving relevant contexts.

We formulate the query augmentation task as an RL problem, where the state is the user's original query $q$, the action is the augmented query $q'$ generated by the model, and the reward is the retrieval performance metric (e.g., recall, NDCG, execution accuracy) achieved when using $q'$ to retrieve contexts.

### 2.2 Reasoning-Enhanced Reinforcement Learning Framework for Information Retrieval

DeepRetrieval employs RL to train language models for query augmentation without requiring the supervision of generation. The key insight is to use actual retrieval performance as the reward signal, allowing the model to learn effective query formulation strategies through trial and error.

Our approach is agnostic to the specific RL algorithm and can be implemented with various policy optimization methods such as PPO (Schulman et al., 2017b) or more recent approaches like GRPO (Guo et al., 2025). We implement DeepRetrieval using the HybridFlow (verl) framework (Sheng et al., 2024), which provides an efficient infrastructure for RL with LLMs.

Inspired by DeepSeek-R1 (Guo et al., 2025), the model is trained to first generate reasoning steps in a <think> section, followed by the final augmented query in an <answer> section. This structured approach enables explicit chain-of-thought reasoning, allowing the model to consider various aspects of the query and explore different augmentation strategies before committing to a final formulation. The output structure is adapted to each retrieval domain's requirements, including structured boolean expressions for literature search and sparse retrieval, natural language query expansion for dense retrieval, and syntactically correct SQL statements for database searching.

### 2.2.1 Reward Function Design

The reward function is critical for aligning DeepRetrieval with effective information retrieval. Our general reward structure follows:

$$r(q, q') = r_{\text{retrieval}}(q, q') + r_{\text{format}}(q') \tag{1}$$

where $r_{\text{retrieval}}$ captures task-specific retrieval performance, and $r_{\text{format}}$ rewards adherence to the required output structure. In this paper, task-specific retrieval metrics include Recall@K for literature search, rank of the answer span hit for evidence-seeking retrieval, NDCG@K for classic IR benchmarks, and execution accuracy for SQL database queries.

The retrieval reward $r_{\text{retrieval}}$ is computed according to task-specific objectives: $r_{\text{retrieval}}(q, q') = \text{Metric}(\mathcal{R}_{q'}, I_q)$, where $\text{Metric}(\cdot, \cdot)$ is the task-specific evaluation function, $\mathcal{R}_{q'}$ represents the retrieval results obtained using the augmented query, and $I_q$ denotes the available information about the query intent or success criteria of the retrieval, i.e., groundtruth. This formulation is flexible enough to accommodate scenarios with explicit ground truth documents (literature search and classic IR), answer spans (evidence-seeking), or execution accuracy (SQL).

We can then optimize DeepRetrieval with PPO algorithm. Formally, given queries sampled from the distribution of training set $\rho$, the LLM policy $\pi$ is optimized with respect to the reward function $r(q, q')$ with a KL-regularized term:

$$\hat{\pi} = \arg\max_{\pi} \mathbb{E}_{q \sim \rho, q' \sim \pi(\cdot|q)} \left[ r(q, q') - \beta \log \frac{\pi(q' \mid q)}{\pi_{\text{ref}}(q' \mid q)} \right] \tag{2}$$

where $\pi_{\text{ref}}$ is a reference policy, typically referred to the initial model before reinforcement learning begins. $\beta$ is an appropriate KL penalty coefficient that controls the strength of the regularization. The KL-divergence term penalizes large deviations from the reference policy, ensuring stability during RL training.

This approach offers several practical advantages over existing methods (Lin et al., 2020; Ye et al., 2023; Ma et al., 2023; Wang et al., 2024b; Wu et al., 2021; Hsu et al., 2024). By directly optimizing for retrieval metrics rather than minimizing the distance to reference or groundtruth queries, DeepRetrieval focuses on metrics that matter in retrieval tasks. The RL framework encourages exploration of the query space, potentially discovering augmentation patterns that human experts might not consider. Additionally, by simply changing the reward function, DeepRetrieval adapts to diverse retrieval scenarios without requiring supervised data for query generation. Finally, DeepRetrieval eliminates the need for expensive human supervision or distillation from larger models, making it more cost-effective for real-world applications.

## 3 Experiments

We evaluate DeepRetrieval on: (1) literature search, (2) evidence-seeking retrieval, (3) classic sparse retrieval, (4) classic dense retrieval, and (5) SQL database search. We summarize the definitions of these retrieval tasks, their evaluation metrics, and reward design in Table 8 in Appendix F. Details of the datasets are presented in Appendix C. Implementation details and software/hardware settings are placed in Appendix B.2 and B.3, respectively.

### 3.1 Literature Search and Evidence-Seeking Retrieval

Table 1 presents our performance on two IR tasks. The first task, literature search, evaluates DeepRetrieval's ability to retrieve relevant documents through a search engine given a specific query. Following Wang et al. (2025), we measure performance using Recall@3K, which represents the percentage of ground truth documents successfully retrieved within the top 3,000 results.[2] The second task, evidence-seeking retrieval, assesses DeepRetrieval's

---

[2]For the literature search task, we use the search APIs provided by https://pubmed.ncbi.nlm.nih.gov/ for publication search , and by https://clinicaltrials.gov/ for trial search.

| | Literature Search | | Evidence-Seeking Retrieval | | | | | | | | |
|---|---|---|---|---|---|---|---|---|---|---|---|
| | **Publication** | **ClinicalTrials** | | **NQ** | | | **TriviaQA** | | | **SQuAD** | |
| | Recall | Recall | H@1 | H@5 | H@20 | H@1 | H@5 | H@20 | H@1 | H@5 | H@20 |
| Original Query | 10.36 | 18.01 | 21.9 | 43.8 | 63.0 | 48.2 | 66.3 | 76.4 | **36.5** | **57.4** | **71.1** |
| GPT-3.5 | 11.67 | 9.42 | 24.3 | 46.0 | 63.9 | 45.8 | 64.3 | 74.2 | 31.6 | 52.4 | 66.6 |
| w/o reasoning | 18.68 | 13.94 | 25.2 | 47.5 | 66.3 | 47.5 | 66.8 | 76.7 | 33.9 | 54.9 | 69.5 |
| GPT-4o | 17.59 | 16.25 | **35.8** | **57.5** | 72.2 | 59.6 | 73.3 | 80.5 | 30.4 | 49.9 | 64.4 |
| w/o reasoning | 19.72 | 14.26 | 29.1 | 56.2 | 69.3 | 53.4 | 70.1 | 78.7 | 33.0 | 52.2 | 66.7 |
| Claude-3-Haiku | 11.26 | 10.10 | 26.2 | 48.6 | 66.4 | 48.8 | 67.9 | 77.7 | 33.3 | 54.1 | 68.4 |
| w/o reasoning | 20.92 | 24.68 | 25.0 | 48.1 | 65.5 | 49.0 | 67.7 | 77.3 | 33.2 | 54.3 | 68.8 |
| Claude-3.5-Sonnet | 20.94 | 18.33 | 35.7 | **57.1** | **72.5** | 57.1 | 71.7 | 79.7 | 28.5 | 48.1 | 63.5 |
| w/o reasoning | 19.08 | 18.41 | **37.2** | 56.9 | **72.7** | **60.8** | **73.8** | **80.6** | 30.3 | 49.8 | 64.7 |
| Mistral[7B-Inst] | 7.18 | 8.08 | 26.9 | 48.8 | 66.0 | 50.0 | 66.7 | 75.9 | 27.7 | 46.6 | 61.6 |
| LEADS[7B] (SFT) | **24.68** | **32.11** | - | - | - | - | - | - | - | - | - |
| Qwen2.5[3B-Inst] | 6.59 | 6.09 | 25.0 | 45.8 | 63.4 | 44.4 | 61.2 | 70.9 | 28.4 | 46.4 | 61.3 |
| w/o reasoning | 9.46 | 7.97 | 23.8 | 45.3 | 64.0 | 46.0 | 64.4 | 74.2 | 32.3 | 52.8 | 66.8 |
| DeepRetrieval[3B] | **65.07** | **63.18** | 35.5 | **57.5** | **72.7** | 58.4 | 73.2 | **80.6** | **38.5** | **59.4** | **72.9** |
| w/o reasoning | **51.90** | **53.31** | 26.9 | 48.8 | 66.9 | 52.0 | 69.4 | 77.7 | 37.8 | 58.0 | 72.5 |

- **[w/o reasoning]** Removes the thinking process (<think>...</think>), forcing the model to generate queries directly.
- LEADS (Wang et al., 2025) is an SFT-based method which trains models with expensive human-annotated and GPT-4o-distilled reference queries. Thus, we only show its performance on literature search tasks, using its original training data.

Table 1: **Performance on literature search with search engines and evidence-seeking retrieval.** We evaluate performance across multiple datasets. For Publication and CinicalTrials, we use Recall@3K as the metric (Wang et al., 2025), with the PubMed and ClinicalTrials.gov search engines (w/ API call) serving as retrievers, respectively. For evidence-seeking retrieval (Natural Questions (NQ), TriviaQA, and SQuAD), we assess performance by measuring "retrieval accuracy" - H@N (answer span hits) (Lin et al., 2021; Ma et al., 2021), using BM25 as the base retriever. We highlight **top-3** and **top-1** performance for each metric.

ability to retrieve documents containing answer spans that match the ground truth answers for a given query. For this evaluation, we follow the methodology of Ma et al. (2021) and use their "retrieval accuracy" H@N (Hits@N) as our metric, which indicates whether at least one document containing an answer span is ranked within the top N retrieved documents.

Notably, on the literature search task, DeepRetrieval achieves remarkable performance with 65.07% and 63.18% recall for publication and trial search, respectively. These results *significantly surpass the previous SOTA performance* achieved by LEADS (Wang et al., 2025), which distilled reference "golden queries" from GPT-4o with human annotation, and used them to fine-tune a Mistral-7B model. This demonstrates the effectiveness of DeepRetrieval in exploring optimal query generation strategies, which we will discuss in detail in §4. Upon examining the reasoning chain and query generation lengths in Figure 4, we discover that the "think" process is crucial for enhancing literature search capability. *Without the "think" process, the model tends to indiscriminately expand the query* in an attempt to find optimal solutions, ultimately making the outcome difficult to optimize.

Moreover, on the evidence-seeking retrieval tasks, DeepRetrieval showcases query rewriting capabilities equivalent to industry-leading LLMs like GPT-4o and Claude-3.5 on NQ and TriviaQA, and significantly outperforms them on SQuAD, despite having only 3B parameters, showing its exceptional parameter efficiency and strong generalization capacity. We analyze this further in §4, with a knowledge injection study presented in Figure 3.

## 3.2 Classic Sparse & Dense Text Retrieval

We further compare classical sparse and dense retrievers under a unified setup across diverse datasets. Table 2 presents the performance of query rewriting models with BM25 as the sparse retriever and E5-Large (Wang et al., 2024a), BGE (Xiao et al., 2024), and Contriever (Izacard et al., 2022) as dense retrievers.

We first observe that DeepRetrieval brings consistent and substantial performance gains across both sparse and dense base retrievers. On the sparse retriever side (BM25), DeepRetrieval-3B demonstrates substantial improvements over all baseline methods on 7 out of 8 datasets. In dense retrieval settings, DeepRetrieval-3B continues to lead, achieving

| Methods | NFCorpus | | FEVER | | HotpotQA | | SciFact | | MS-Beir | | MS-H | | MS-S | | MS-T | |
|---|---|---|---|---|---|---|---|---|---|---|---|---|---|---|---|---|
| | S | D | S | D | S | D | S | D | S | D | S | D | S | D | S | D |
| **Base Retriever** | | | | | | | | | | | | | | | | |
| BM25 / Dense | 14.7 | 37.0 | 44.2 | 82.5 | 61.1 | 70.0 | 57.3 | 64.5 | 44.8 | 70.4 | 32.5 | 32.4 | 38.8 | 31.1 | 51.3 | 49.8 |
| **Zero-shot Query Gen (w/o reasoning)** | | | | | | | | | | | | | | | | |
| GPT-3.5 | 30.1 | 33.0 | 55.0 | 64.5 | 58.1 | 54.1 | 66.4 | 58.9 | 43.1 | 69.1 | 28.8 | 31.7 | 35.4 | 33.0 | 48.8 | 50.6 |
| GPT-4o | 31.8 | 33.6 | 59.1 | 72.2 | 58.0 | 70.2 | 66.4 | 65.5 | 47.8 | 68.5 | 21.8 | 27.8 | 28.5 | 28.5 | 43.4 | 48.2 |
| Claude-3-Haiku | 31.3 | 26.6 | 43.5 | 73.6 | 49.3 | 62.1 | 65.1 | 63.0 | 38.7 | 69.0 | 29.0 | 31.1 | 34.7 | 33.5 | 48.9 | 52.0 |
| Claude-3.5-Sonnet | 31.6 | 35.2 | 54.8 | 71.0 | 46.4 | 58.7 | 68.4 | **68.2** | 45.3 | 61.1 | 21.3 | 21.9 | 27.5 | 24.2 | 39.7 | 43.7 |
| Qwen2.5-3B-Inst | 20.9 | 33.9 | 55.5 | 71.4 | 51.7 | 64.7 | 65.1 | 62.9 | 31.3 | 66.9 | 26.3 | 30.5 | 31.6 | 33.0 | 46.7 | 49.2 |
| **Zero-shot Query Gen (w/ reasoning)** | | | | | | | | | | | | | | | | |
| GPT-3.5 | 32.1 | 32.8 | 55.4 | 63.9 | 54.4 | 54.1 | 65.1 | 61.9 | 39.3 | 63.1 | 20.8 | 28.8 | 25.7 | 30.0 | 40.5 | 47.6 |
| GPT-4o | 30.7 | 34.0 | 53.9 | 73.8 | 56.4 | **71.8** | 65.0 | 63.8 | 39.4 | 66.6 | 20.8 | 20.6 | 26.4 | 23.0 | 42.0 | 44.2 |
| Claude-3-Haiku | 29.6 | 36.0 | 59.9 | 74.9 | 55.6 | 65.2 | **68.9** | 65.6 | 44.7 | 67.2 | 16.5 | 24.2 | 22.4 | 25.4 | 37.6 | 43.5 |
| Claude-3.5-Sonnet | 30.7 | 36.4 | 55.7 | 75.8 | 55.2 | 67.6 | 68.7 | 65.9 | 47.8 | 63.9 | 18.6 | 18.5 | 27.3 | 23.6 | 41.6 | 43.8 |
| Qwen2.5-3B-Inst | 29.2 | 32.4 | 46.7 | 69.9 | 48.3 | 62.0 | 63.8 | 62.1 | 23.0 | 60.0 | 22.7 | 24.3 | 25.9 | 27.6 | 43.0 | 45.0 |
| **Ours (DeepRetrieval-3B)** | | | | | | | | | | | | | | | | |
| +BM25 / +Dense | **34.0** | **37.7** | **66.4** | **84.1** | **63.1** | 70.1 | 64.6 | 66.4 | **53.1** | **70.4** | **34.7** | **32.5** | **41.1** | **36.1** | **53.8** | **52.3** |

Table 2: **NDCG@10 Performance of Classic Sparse (S) and Dense (D) Document Retrieval.** We use BM25 as the base sparse retriever for all the datasets, while using E5-Large as the base dense retriever for SciFact, use BGE-base-en-v1.5 for HotpotQA, FEVER, NFCorpus, and MS-Beir, and use vanilla Contriever for MS MARCO domain-specific (MS-H: health, MS-S: science, MS-T: technology) subsets. The best performance score is denoted in **bold**.

the best performance on 6 out of 8 datasets. These results demonstrate that DeepRetrieval, trained via RL, is not only competitive with, but often significantly outperforms, strong commercial LLMs such as GPT-4o and Claude-3.5 on classic text retrieval benchmarks. The consistent gains across both sparse and dense base retrievers highlight **the retriever-agnostic nature of our framework**. By leveraging RL, DeepRetrieval model learns to generate queries that align with the preferences of the underlying retriever, leading to substantial improvements in retrieval performance regardless of the types of base retrievers.

We further observe that on datasets such as HotpotQA, FEVER, and MS MARCO (MS-Beir), the dense retriever BGE-base-en-v1.5 achieves remarkably strong NDCG@10 scores—often exceeding 0.7. In these cases, the performance improvements from applying DeepRetrieval are relatively small. Upon closer examination, we find that BGE-base-en-v1.5 was trained using data from the training splits of HotpotQA, FEVER, and MS MARCO (FlagOpen, 2023). This prior exposure suggests that for these datasets, the retriever is already near-optimal, leaving limited room for improvement through query rewriting. This observation also explains why many other query augmentation models lead to decreased performance when used with BGE. The queries generated by these models often deviate from the patterns BGE was exposed to during pretraining. As a result, even though the augmented queries might seem more informative, they can harm retrieval performance because they no longer align with the retriever's learned preference.

To further investigate this phenomenon, we conduct additional experiments on three MS MARCO subsets using vanilla Contriever, a dense retriever that has not been trained on MS MARCO unlike BGE. When used alone, Contriever yields relatively low retrieval performance on MS-H and MS-S, with NDCG@10 scores around 30%. However, after applying DeepRetrieval, we observe substantial improvements—a nearly 5-point gain on MS-S (D). This indicates that DeepRetrieval is especially effective for unadapted retrievers, which have not achieved near-optimal performance on a particular dataset's query-document distribution. Interestingly, on the MS-H dataset, DeepRetrieval does not yield as significant an improvement as observed on MS-S and MS-T. We hypothesize that this is due to the lack of domain-specific knowledge in our base model (Qwen-2.5-3B-Instruct), or that the dense retriever has already reached its performance ceiling on this domain, limiting the impact of query augmentation. Investigating the role of domain adaptation in both the query rewriting model and the retriever within the DeepRetrieval framework is a promising direction for future work. DeepRetrieval is not limited to unadapted retrievers. Even when applied to high-performing retrievers like BGE, which may have seen training-split data of the evaluation datasets, our method still provides gains. For example, on FEVER, DeepRetrieval improves performance from 82.5 to 84.1 (+1.6), demonstrating its ability to enhance already strong retrievers. This highlights the general applicability of our framework across dense retrieval models.

Notably, we also find that BM25 combined with DeepRetrieval can match or even outperform dense retrievers with DeepRetrieval on several datasets. For instance, on NFCorpus and SciFact, the performance of BM25+DeepRetrieval is close to Dense+DeepRetrieval. More surprisingly, on the MS MARCO domain-specific subsets, BM25+DeepRetrieval actually surpasses dense retrieval in terms of NDCG@10. This is a notable result, especially when considering the significant efficiency advantage of sparse retrieval methods like BM25 over dense retrieval. To illustrate the efficiency of BM25, we compare the total runtime—including both query generation and document retrieval—on a corpus of 5.42 million documents and 13,332 queries. Under identical settings except for the retrieval method, dense retrieval required 12,232 seconds to complete, while BM25 finished in only 352 seconds. Given that query generation time is nearly close in both cases, this 34× speedup reflects the significantly faster performance of BM25 in the retrieval phase. This opens up a promising future direction: using DeepRetrieval to push BM25 to a new height in retrieval quality, effectively combining high performance with high efficiency. In other words, our approach shows potential to make BM25 great again, and we leave a deeper exploration of this direction to future work.

### 3.3 SQL Database Search

SQL database search is another form of information retrieval (Baeza-Yates et al., 1999), and we aim to explore how our approach can be extended to text-to-SQL query generation (Deng et al., 2022) for database search. Table 3 presents the performance of various base models trained with different methods on two text-to-SQL datasets for database search.

The results show that DeepRetrieval consistently improves performance compared to the base model. With RL from scratch, execution accuracy improves by 18.25% on BIRD and 23.88% on Spider compared to Qwen2.5-Coder$_{\text{3B-Inst}}$ in a zero-shot setting. Despite having only 3 billion parameters, it outperforms GPT-4o and Claude-3.5 on the BIRD dataset. SFT models are trained on high-quality, human-curated SQL with reasoning processes distilled from GPT-4o. However, models trained with RL from scratch can even outperform such SFT models without using groundtruth SQL queries (e.g., DeepRetrieval$_{\text{3B-Base}}$ vs Qwen2.5$_{\text{3B-Inst}}$ (SFT) and DeepRetrieval$_{\text{3B-Coder}}$ vs Qwen2.5-Coder$_{\text{3B-Inst}}$ (SFT)). This shows that using execution accuracy as a direct reward is more effective for SQL database search tasks than SFT with groundtruth SQL. We hypothesize that although humans provide high-quality SQL, it is not necessarily optimal. Through RL, LLMs can explore the search space to discover diverse SQL generation strategies, instead of "memorizing" some suboptimal ones.

| Methods | BIRD | Spider |
|---|---|---|
| **Zero-shot (w/o reasoning)** | | |
| GPT-3.5 | 46.22 | 67.02 |
| GPT-4o | 55.35 | 73.50 |
| Claude-3-Haiku | 43.16 | 64.88 |
| Claude-3.5-Sonnet | 50.46 | 60.74 |
| Qwen2.5$_{\text{3B-Inst}}$ | 29.66 | 52.90 |
| Qwen2.5-Coder$_{\text{3B-Inst}}$ | 30.77 | 50.97 |
| Qwen2.5-Coder$_{\text{7B-Inst}}$ | 45.24 | 64.89 |
| **Zero-shot (w/ reasoning)** | | |
| GPT-3.5 | 44.07 | 64.88 |
| GPT-4o | 55.93 | 73.40 |
| Claude-3-Haiku | 43.81 | 67.44 |
| Claude-3.5-Sonnet | 50.65 | 66.05 |
| Qwen2.5$_{\text{3B-Inst}}$ | 30.83 | 55.13 |
| Qwen2.5-Coder$_{\text{3B-Inst}}$ | 33.57 | 54.45 |
| Qwen2.5-Coder$_{\text{7B-Inst}}$ | 45.57 | 67.70 |
| **SFT (w/o reasoning)** | | |
| Qwen2.5$_{\text{3B-Inst}}$ | 33.77 | 56.67 |
| Qwen2.5-Coder$_{\text{3B-Inst}}$ | 39.77 | 58.61 |
| Qwen2.5-Coder$_{\text{7B-Inst}}$ | 44.07 | 65.96 |
| **SFT (w/ reasoning)** | | |
| Qwen2.5$_{\text{3B-Inst}}$ | 37.29 | 60.93 |
| Qwen2.5-Coder$_{\text{3B-Inst}}$ | 46.15 | 66.92 |
| Qwen2.5-Coder$_{\text{7B-Inst}}$ | 50.65 | 70.99 |
| **Ours** | | |
| DeepRetrieval$_{\text{3B-Base}}$ | 41.40 | 68.79 |
| w/ cold start | 44.00 | 70.33 |
| w/o reasoning | 39.57 | 70.24 |
| DeepRetrieval$_{\text{3B-Coder}}$ | 49.02 | 74.85 |
| w/ cold start | 50.52 | 74.34 |
| w/o reasoning | 47.00 | 73.59 |
| DeepRetrieval$_{\text{7B-Coder}}$ | **56.00** | **76.01** |

Table 3: **Execution Accuracy of SQL Database Search on BIRD and Spider.** "w/ cold start" (Guo et al., 2025) means that we start RL with the SFT model.

## 4 Discussions & Takeaways

**Knowledge Injection Matters and DeepRetrieval Can Adaptively Learn It.** Our analysis of evidence-seeking retrieval tasks reveals that LLMs may incorporate answer information into generated queries based on prior knowledge. As shown in Figure 3, the contribution

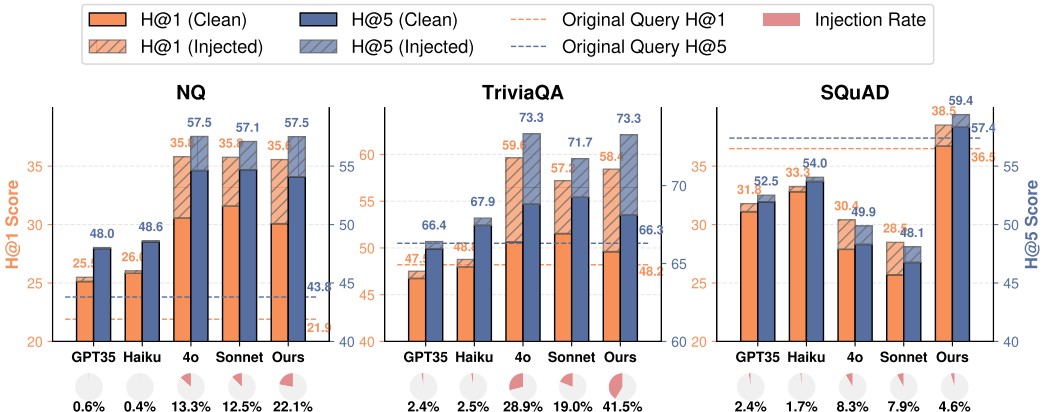

Figure 3: **Knowledge Injection Study on Evidence-Seeking Retrieval**. LLMs may input answer information to the generated queries based on their prior knowledge. We implement a postprocessing method (Appendix E) where an auxiliary LLM analyzes the query, identifies knowledge injections, and removes the injections to see the contribution of this factor.

of knowledge injection varies significantly across datasets. For NQ, performance gains stem from both knowledge injection and other factors (e.g., query standardization), with performance remaining strong even after removing injected knowledge. In contrast, TriviaQA improvements are predominantly attributed to knowledge injection, with performance dropping significantly when injected content is removed. For SQuAD, knowledge injection appears to have minimal impact despite relatively high injection rates from GPT-4o and Claude-3.5-Sonnet, suggesting success depends more on understanding dataset distribution than prior knowledge. DeepRetrieval demonstrates *adaptive behavior* across these datasets, applying moderate (22.1%) knowledge injection for NQ, extensive (41.5%) injection for TriviaQA, and minimal injection (4.6%) for SQuAD. This suggests the model effectively learns optimal augmentation strategies for each dataset type, explaining its consistent performance advantages across diverse retrieval tasks.

**Direct Optimization vs Mimicry: Why RL Surpasses SFT in Search Tasks?** Our experimental results demonstrate the superiority of reinforcement learning (RL) over supervised fine-tuning (SFT) across different retrieval scenarios. For SQL database search in Table 3, we observe that RL from scratch achieves better performance than SFT, no matter the latter has reasoning chain or not. The contrast is even more pronounced in literature search in Table 1, where RL from scratch substantially outperforms SFT approaches like LEADS (Wang et al., 2025), which is finetuned on expensive GPT-4o-distilled, human-annotated "golden" queries. This RL-based approach is particularly effective for real search engines, where the ability to formulate precise boolean expressions with appropriate operator usage and term grouping is crucial. Unlike supervised methods which are constrained by potentially suboptimal reference queries, DeepRetrieval's RL framework directly optimizes for the search engines' ranking behavior, discovering query patterns that might not be intuitive to human experts but work exceptionally well with the systems' underlying algorithms. This pattern suggests that RL is universally applicable across different retrieval scenarios, with the gap between RL and SFT varying based on the availability of definitive ground truth (as in SQL tasks) versus more ambiguous contexts. In latter cases, RL's ability to explore the solution space through trial and error leads to superior outcomes compared to mimicking reference solutions. This aligns with recent findings by Chu et al. (2025), who demonstrate that while SFT tends to memorize patterns from training data, RL develops more generalizable strategies by focusing on optimizing task-specific rewards. The cold start results in Table 3 results further indicate that SFT can provide a strong initialization for RL especially when the LLM lacks a certain capability (e.g., DeepRetrieval$_{3B-Base}$ has limited coding capability compared to DeepRetrieval$_{3B-Coder}$), combining the benefits of both approaches when quality supervision is available.

**Why Reasoning Process Can Particularly Improve DeepRetrieval in RL Training?** Our results in Table 1 demonstrate that incorporating a reasoning process before generating the final answer in DeepRetrieval RL training significantly enhances task performance. This improvement can be attributed to the fact that reasoning encourages broader exploration during training. Specifically, requiring the model to generate a reasoning process before the final answer encourages more diverse and contextually enriched queries. This is because the reasoning step prompts the LLM to generate additional information, which in turn enhances the quality of the generated answer. By explicitly reasoning over the given user query, the model explores multiple semantic interpretations, leading to a broader range of candidate queries. This process enables the model to better learn the preferences of the search engine—i.e., how to formulate queries that yield optimal retrieval results. Consequently, the search engine retrieves a more comprehensive and relevant set of documents, leading to higher recall. We can observe a similar phenomenon in Figure 4(c), where the Qwen model without reasoning ("Qwen (No Think)") underperforms the version with reasoning throughout the training steps. Furthermore, we observe an intriguing trend: the length of the reasoning chains decreases over time. This is in contrast to the "aha moment" phenomenon observed in DeepSeek-R1 (Guo et al., 2025), where the reasoning chains became increasingly longer as training progressed. A plausible explanation for this is, unlike tasks that demand explicit multi-step reasoning, such as math problems and coding, our retrieval-focused setup does not inherently need long-form reasoning. *Reasoning was not essential to solving the task itself, but rather served as an auxiliary tool for exploration*. Once exploration helped the model discover a performant query generation strategy, the need for reasoning diminished—resulting in the observed convergence toward shorter reasoning chains. We discuss more (e.g., why Qwen without thinking increases the query lengths) in Appendix D.1.

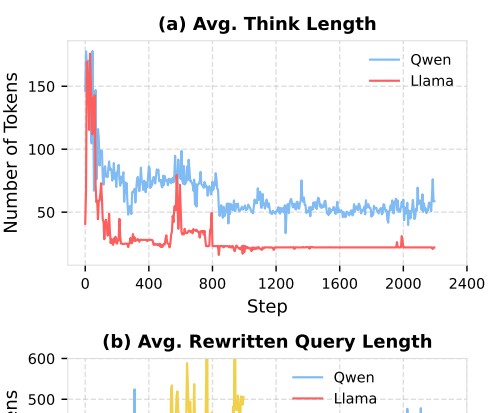

(a) Avg. Think Length

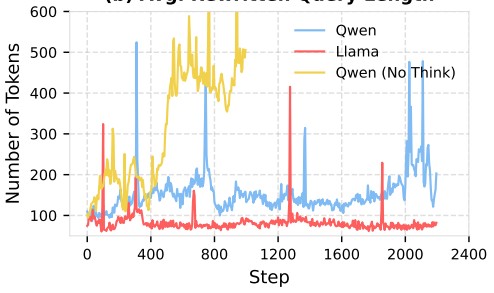

(b) Avg. Rewritten Query Length

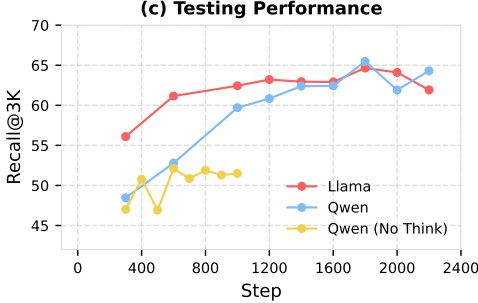

(c) Testing Performance

Qwen (No Think) suffered from excessive decoding time, making training and testing over 8× slower than Qwen and Llama. For instance, testing at Step 1000 took ~10.5 hours, compared to just 1 hour for models with <think>. Thus, we halted the training early.

Figure 4: **Thinking (Reasoning Chain) and Query Generation Length Study on Literature Search via Search Engine (PubMed).** We test the following settings of DeepRetrieval: (1) with Qwen-2.5-3B-Inst as the base model, (2) with Llama-3.2-3B-Inst as the base model, and (3) setting 1 without thinking.

## 5 Conclusion

We introduced DeepRetrieval, an RL-based approach for query generation that optimizes retrieval metrics without supervised reference queries. DeepRetrieval outperforms prior methods across diverse tasks, doubling recall in literature search, rivaling industry-leading models in evidence-seeking and information retrieval with just 3B parameters, and excelling in SQL search. These results highlight RL as a more effective paradigm for information retrieval, transforming how users connect with information. We comprehensively discuss the related works in Appendix A.

## Acknowledgment

Research was supported in part by National Science Foundation IIS-19-56151, the Molecule Maker Lab Institute: An AI Research Institutes program supported by NSF under Award No. 2019897, and the Institute for Geospatial Understanding through an Integrative Discovery Environment (I-GUIDE) by NSF under Award No. 2118329, US DARPA INCAS Program No. HR0011-21-C0165 and BRIES Program No. HR0011-24-3-0325.

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

**Contents of Appendix**

# A Related Work

## A.1 Query Augmentation for Search Engines

Poorly written queries fail to fully capture the required information (Belkin et al., 1982), which has become a major issue in various IR systems (Schwartz, 1998; Dageville et al., 2016). To tackle this problem, query augmentation (Rocchio Jr, 1971) was proposed to refine the original query or enrich it with additional information. Its development can be divided into two phases, marked by the advent of LLMs.

**Classical Query Augmentation/Generation** Prior to the introduction of LLMs (Vaswani et al., 2017; Devlin et al., 2019), query augmentation methods largely fell into two categories. Pseudo-Relevance Feedback (PRF) (Rocchio Jr, 1971; Zhai & Lafferty, 2001; Abdul-Jaleel et al., 2004; Metzler & Croft, 2005; 2007) assumes that the top-k documents retrieved initially are relevant, and leverages them to enrich the original query. With the development of knowledge bases (Zesch et al., 2007; 2008; Syed, 2010), knowledge base-driven query augmentation (Dalton et al., 2014; Xu et al., 2009; Meij et al., 2010; Xiong & Callan, 2015) has also shown promising performance by using external knowledge to refine user queries. Text-to-SQL (Deng et al., 2022; Gao et al., 2023b; Hong et al., 2025) is another form of query augmentation, where the input is a user query and the output is the SQL used for database search. Previous methods (Zhong et al., 2017) typically focus on training language models on high-quality SQL or designing deliberate prompts or pipelines for them (Pourreza & Rafiei, 2023; Li et al., 2024).

**LLM-based Query Augmentation** The advent of LLMs has brought significant progress to the field of query augmentation. Leveraging LLMs (Khattab & Zaharia, 2020; Wang et al., 2022a) as dense retrieval (Karpukhin et al., 2020) engines enables more accurate PRF (Yu et al., 2021; Wang et al., 2021; 2022b). Furthermore, LLMs (Brown et al., 2020; Touvron et al., 2023) are also directly used to refine queries by rewriting content (Lin et al., 2020; Yu et al., 2020; Ye et al., 2023), adding background knowledge (Gao et al., 2022; Liu et al., 2022; Mackie et al., 2023) or providing intermediate explanation (Pereira et al., 2022; Ferraretto et al., 2023; Jagerman et al., 2023).

## A.2 Reinforcement Learning and LLMs in Query Augmentation

**RL for General LLM Training** Reinforcement Learning (RL) (Kaelbling et al., 1996) is an ML training strategy that teaches agents to make decisions via interacting with the environment and maximizing rewards. Recently, RL attracted significant attention in the era of LLMs, serving as a powerful framework for aligning models with human preferences. One representative work is Reinforcement Learning from Human Feedback (RLHF) (Christiano et al., 2017; Stiennon et al., 2020; Ouyang et al., 2022) which uses the PPO (Schulman et al., 2017a) and human preference data to train a reward model that guides the optimization of LLMs. To reduce the complexity of multi-round interactions between LLMs and the environment, simplified alternatives such as Direct Preference Optimization (DPO)(Rafailov et al., 2023) and SimPO (Meng et al., 2024) have been proposed. Furthermore, algorithms like GRPO (Guo et al., 2025), REINFORCE++ (Hu, 2025) are introduced to improve reward modeling and avoid biased optimization (Xu et al., 2024).

**RL for LLM-based Query Augmentation** Given the success in LLM post-training (Kumar et al., 2025), RL training strategy has also been explored in IR domain. For query augmentation, RL strategies can be classified into three types based on their reward design.

- **Reward from Retrieval Metrics** A direct method for designing query rewriting rewards involves leveraging retrieval metrics, such as recall and NDCG. For instance, Wu et al. (2021) constructed a preference dataset by approximating retrieval metrics, supervised by ground-truth document labels, to fine-tune a T5 model (Raffel et al., 2019) for conversational query rewriting. Furthermore, Hsu et al. (2024) created a preference dataset to optimize Llama 3 (Grattafiori et al., 2024) and Gemma (Team et al., 2024) to diversify user

queries. They employed IPO (Azar et al., 2023) and context distillation (Snell et al., 2022), utilizing retrieval metrics rewards guided by ground-truth document labels.

- **Reward from Generator Performance (Reward Models)** The second approach designs rewards based on the generator's performance in a Retrieval-Augmented Generation (RAG) (Fan et al., 2024; Zhao et al., 2024) framework. It uses the quality of the generator's outputs rather than directly evaluating the rewrite itself as a reward to guide optimization. Ma et al. (2023) first train a T5 (Raffel et al., 2019)-based query rewriter through supervised fine-tuning as a warm-up phase, then apply PPO (Schulman et al., 2017a) optimization. In this approach, the reward is determined by the downstream generator's performance within the RAG framework.

- **Reward from Both** The third reward design also appears in RAG frameworks, combining elements from the previous two methods. Wang et al. (2024b) perform supervised warm-up followed by PPO optimization. They utilize reward signals derived from the similarity among rewritten queries, retrieved documents, and human-annotated ground truth, along with the ROUGE (Lin, 2004) scores of the downstream generator outputs. Using these reward signals, they further train three LLMs respectively as reward models.

### A.3 Difference between DeepRetrieval and Previous Work

In this section, we highlight the key differences between DeepRetrieval and prior LLM-based query augmentation methods.

- **No Supervision for Query Generation** Unlike previous approaches that rely on supervision for query generation as a warm-up phase (Ma et al., 2023; Wang et al., 2024b) or as the main optimization strategy (Liu et al., 2022; Ye et al., 2023; Wang et al., 2025) or few-shot examples (Yu et al., 2020)—and thus require costly human-annotated query rewrites or computational resources, our method eliminates the need for any such supervision.

- **RL with Trial & Error** Unlike previous methods (Wu et al., 2021; Hsu et al., 2024) that adopt RL strategies similar to DPO (Rafailov et al., 2023) by fine-tuning on static preference datasets, DeepRetrieval uses PPO (Schulman et al., 2017a) as the primary training algorithm. This enables on-policy, dynamic trial-and-error during training, leading to more substantial performance improvements.

- **RL with Robust Reward Function** In designing the reward function, DeepRetrieval advances the philosophy of using retrieval metrics as reward signals. It directly transforms recall, NDCG, answer spans, and execution accuracy into reward signals across the tasks of IR search, evidence seeking, and SQL generation. Compared to methods that use reward from downstream generator performance, i.e., reward modes (Ma et al., 2023; Wang et al., 2024b), DeepRetrieval is more direct and robust. Compared to prior approaches that derive rewards from retrieval metrics (Wu et al., 2021; Hsu et al., 2024), DeepRetrieval provides a more comprehensive and flexible design for reward signals.

With the features above, DeepRetrieval achieves dominant performance across a range of tasks without relying on any supervision for query generation. Remarkably, with just a 3B-parameter open-source model (Yang et al., 2024), DeepRetrieval outperforms commercial models like GPT-4o (Hurst et al., 2024) and Claude-3.5-Sonnet (Anthropic, 2024), demonstrating exceptional efficiency and capability of our methodology.

## B Theoretical Foundations & Implementation Details

### B.1 Additional Details of PPO Algorithms

DeepRetrieval can be implemented with various policy optimization algorithms. Here we provide details on our implementation with PPO, while noting that the framework is compatible with other algorithms like GRPO.

**Advantage Estimation and Policy Optimization** To optimize our DeepRetrieval framework using the reward function defined in Eq. 1 and objective function in Eq. 2, we can employ

Proximal Policy Optimization (PPO) (Schulman et al., 2017b) with Generalized Advantage Estimation (GAE) (Schulman et al., 2015). This approach enables stable and efficient learning while directly optimizing for improved retrieval performance.

The core idea is to iteratively update our query augmentation policy to maximize the expected reward while avoiding destructively large updates. For each original query $q$, our policy $\pi_\theta$ generates an augmented query $q'$, receives retrieval-based rewards, and then updates its parameters to generate better queries in the future.

Our PPO objective is formulated as:

$$L^{\text{PPO}}(\theta) = L^{\text{CLIP}}(\theta) - c_1 L^{\text{VF}}(\theta) + c_2 S[\pi_\theta] \tag{3}$$

The first term, $L^{\text{CLIP}}$, is the core learning objective that uses clipping to ensure stable policy updates:

$$L^{\text{CLIP}}(\theta) = \mathbb{E}_t[\min(r_t(\theta)A_t, \text{clip}(r_t(\theta), 1 - \epsilon, 1 + \epsilon)A_t)] \tag{4}$$

Here, $r_t(\theta) = \frac{\pi_\theta(q'_t|q_t)}{\pi_{\theta_{\text{old}}}(q'_t|q_t)}$ represents how much the current policy differs from the previous one when generating augmented query $q'_t$ from original query $q_t$. The advantage term $A_t$ indicates how much better (or worse) the augmented query performed compared to what we expected, based directly on the retrieval rewards defined in Eq. 1. The clipping operation prevents extreme policy changes by limiting updates to the range $[1 - \epsilon, 1 + \epsilon]$. The second term, $L^{\text{VF}}$, trains a value function that predicts expected rewards, helping to reduce variance in training. The third term, $S[\pi_\theta]$, adds entropy bonus to encourage exploration of diverse query formulations, which is particularly important for discovering novel and effective query augmentation strategies. For advantage estimation, we leverage GAE (Schulman et al., 2015), which provides a good balance between bias and variance in our advantage estimates, which allows for more stable and efficient learning from our retrieval-based rewards. This optimization framework directly connects our policy updates to improvements in retrieval performance metrics (e.g., Recall, NDCG, execution accuracy) specified in our reward functions, ensuring that DeepRetrieval progressively learns more effective query augmentation strategies.

**Training Process and Practical Advantages** Our training process consists of three main stages per iteration. (1) In the Generation Stage, the actor model produces augmented queries from original queries. (2) The Preparation Stage follows, where the critic computes values, and the reward model computes rewards based on retrieval performance. (3) Finally, in the Learning Stage, the actor and critic models are updated using the chosen RL algorithm. For efficient training, we utilize HybridFlow (verl)'s (Sheng et al., 2024) distributed training capabilities with tensor parallelism during training and hybrid data-model parallelism during inference.

## B.2  PPO Implementation

We train our query-generation LLM using the PPO algorithm, with the Qwen2.5-3B-Instruct model as both the actor and the critic. The actor is trained with a learning rate of 1e-6, while the critic uses a slightly larger learning rate of 1e-5 to allow faster value approximation. The KL coefficient is set to 0.001, balancing exploration and policy stability. The generation temperature is set to 0.6, encouraging a mix of determinism and diversity in generated reasoning chains and queries. The batch size is set as 64 and the PPO mini batch size is 16. In the context of document retrieval, when computing rewards based on NDCG@$k$, we select a larger value of $k$ (3000 in our case) compared with the evaluation phase, to mitigate the risk of sparse or zero-reward signals. For literature search, we additionally test DeepRetrieval with LLaMA-3.2-3B-Instruct as the base model on the Publication (PubMed) dataset, keeping all the hyperparameters unchanged. For SQL database search, we also test the Qwen2.5-Coder-3B-Instruct and Qwen2.5-Coder-7B-Instruct models to investigate how performance is impacted by the models' inherent coding capabilities. We set the maximum response (new token) lengths for literature search, evidence-seeking retrieval, classic retrieval, and SQL generation to 500, 350, 512, and 512, respectively.

### B.3 Experimental Settings

**Software** We implemented DeepRetrieval using Python 3.9 with the VERL framework (Sheng et al., 2024)[3] as the core architecture for reinforcement learning with language models. Our implementation utilizes VLLM (v0.6.3) (Kwon et al., 2023) for efficient LLM inference and generation, PyTorch (v2.4.0) with CUDA 12.1 support for deep learning operations, and Ray (Moritz et al., 2018) for distributed training and inference. We incorporated Flash Attention 2 (Dao, 2023) for optimized attention calculations and PySerini (v0.22.1) (Lin et al., 2021) for efficient retrieval and evaluation. For efficient dense retrieval, we employed FAISS-GPU (v1.7.2) (Douze et al., 2024). Most experiments were conducted with Qwen2.5-3B-Instruct (Yang et al., 2024) as the base model, with gradient checkpointing enabled when necessary to reduce memory consumption. We additionally tested LLaMA-3.2-3B (Grattafiori et al., 2024) on literature search task, Qwen2.5-Coder-3B-Instruct and Qwen2.5-Coder-7B-Instruct on SQL database search task. Regarding proprietary LLMs, we employ Microsoft Azure[4] to call GPT models and AWS Bedrock[5] to call Claude models. The versions of Claude-3-Haiku and Claude-3.5-Sonnet on AWS Bedrock we used are `anthropic.claude-3-haiku-20240307-v1:0` and `anthropic.claude-3-5-sonnet-20240620-v1:0`, respectively.

**Hardware** For 3B models, all the tasks are run on two NVIDIA A100 80GB PCIe on a system with an AMD EPYC 7513 32-Core Processor and 1.0 TB of RAM. BIRD and Spider with DeepRetrieval$_{Coder-7B}$ are run on four NVIDIA A100 80GB PCIe.

**Other Training details** For the SQL database search task, we train the SFT model for six epochs with a batch size of 8 and a learning rate of 2e-5. The best performance for all the SFT models with reasoning is achieved at the fourth epoch, while the SFT model without reasoning performs best at the second epoch. To avoid over-memorization, we select the SFT model with reasoning from the first epoch as the corresponding cold start model. We also attempted to split the training set ($\sim$8k) to separately conduct SFT for cold start and RL. However, due to the limited data size, the performance of the RL training with these cold start models were not satisfactory.

---

[3]https://github.com/volcengine/verl
[4]https://azure.microsoft.com/
[5]https://aws.amazon.com/bedrock/

## C  Dataset Details

### C.1  Literature Search

The statistics of literature search datasets are presented in Table 4. The datasets are sourced from LEADS (Wang et al., 2025), which were constructed by linking systematic reviews to their cited studies. For publications, systematic reviews from the MS2 dataset were used, and their referenced studies were retrieved via the PubMed API. To ensure comprehensive coverage, additional PubMed citations were linked to clinical trials using NCT IDs when available. For clinical trials, structured records from ClinicalTrials.gov were filtered based on reported results and full-text availability.

| Dataset | # Train Queries | # Val Queries | # Test Queries | Search Engine |
|---|---|---|---|---|
| Publication | 12,801 | 4,217 | 4,217 | PubMed |
| ClinicalTrial | 5,074 | 857 | 1,692 | ClinicalTrials.gov |

Table 4: Statistics of the literature search datasets.

We show some examples of data below.

---

**Query:**
*P:* Women who have experienced perineal trauma following childbirth
*I:* Rectal analgesia for pain relief
*C:* Other forms of pain management such as local anaesthetics, oral analgesics, therapeutic ultrasound, antiseptics, ice packs, baths
*O:* Effectiveness of rectal analgesia in relieving pain from perineal trauma

**Relevant Publications (PMID):** [9647153, 15511762, 2892739]

---

Figure 5: Example of a literature search query and groundtruth publications on PubMed.

---

**Query:**
*P:* Adults with cardiometabolic risk factors
*I:* Supplementation of hydroxycinnamic acids (HCAs) from foods/extracts
*C:* No supplementation or placebo
*O:* Effect on cardiometabolic biomarkers such as cholesterol, blood pressure, and glycaemia

**Relevant Cinical Trials (NCTID):** [NCT01293175, NCT00827450, NCT01596309]

---

Figure 6: Example of a literature search query and groundtruth trials on ClinicalTrials.gov.

PICO is a widely used query format for search in medicine (Schardt et al., 2007; Huang et al., 2006; Eriksen & Frandsen, 2018; Riva et al., 2012) where P, I, C, and O are defined as:

P: Patient, Problem or Population - Who or what is the research about? I: Intervention - What is the main intervention or exposure being considered? C: Comparison - What is the intervention being compared to? O: Outcome - What are the relevant outcomes or effects being measured?

## C.2 Evidence-Seeking Retrieval

The statistics of literature search datasets are presented in Table 5. The datasets we used and the experimental settings are consistent with Ma et al. (2021).

| Dataset | # Train Queries | # Val Queries | # Test Queries | Corpus | Corpus Size |
|---|---|---|---|---|---|
| Natural Questions | 79,168 | 8,757 | 3,610 | wikipedia-100w | 21M |
| TriviaQA | 78,785 | 8,837 | 11,313 | wikipedia-100w | 21M |
| SQuAD | 87,599 | - | 10,570 | wikipedia-100w | 21M |

Table 5: Statistics of the evidence-seeking retrieval datasets. "wikipedia-100w" refers to the English Wikipedia's dump from 2018-12-20 in non-overlapping 100-word splits (Ma et al., 2021; Lin et al., 2021).

We show some examples of data below.

> **Query:** "who is currently the longest serving supreme court justice"
> **Candidate Answers:** ["Anthony Kennedy"]
>
> **Query:** "who did the music for the new blade runner film"
> **Candidate Answers:** ["Hans Zimmer", "Benjamin Wallfisch"]

Figure 7: Data samples in Natural Questions (NQ).

> **Query:** "Who founded the off-Broadway theater where Hair had its premier?"
> **Candidate Answers:** ["Joe Papp", "Joseph Papp", "Joseph Papirofsky"]
>
> **Query:** "What number Star Trek movie was called The Wrath of Khan?"
> **Candidate Answers:** ["II", "2", "two"]

Figure 8: Data samples in TriviaQA.

> **Query:** "Where was Super Bowl 50 held?"
> **Candidate Answers:** ["Santa Clara, California.", "Levi's Stadium", "Levi's Stadium"]
>
> **Query:** "Who led the Broncos with 105 receptions?"
> **Candidate Answers:** ["Demaryius Thomas", "Demaryius Thomas", "Thomas"]

Figure 9: Data samples in SQuAD.

## C.3 Classic Sparse & Dense Text Retrieval

### C.3.1 Statistics of Text Retrieval Datasets from BEIR

We summarize the dataset statistics in Table 7. Five datasets are sourced from the BEIR benchmark (Thakur et al., 2021), covering a diverse range of retrieval scenarios including fact checking (FEVER, SciFact), multi-hop QA (HotpotQA), and domain-specific biomedical retrieval (NFCorpus). For each dataset, we report the number of training, validation, and test queries, as well as the size of the associated document corpus. For SciFact, no official validation split is provided. Thus, we use the data provided in Lin et al. (2024), which randomly select a subset from the training set for validation purposes.

| Dataset | # Train Queries | # Val Queries | # Test Queries | Corpus Size | Source |
|---|---|---|---|---|---|
| NFCorpus | 2,590 | 647 | 323 | 3.6K | BEIR |
| FEVER | 109,810 | 13,332 | 6,666 | 5.42M | BEIR |
| HotPotQA | 85,000 | 12,852 | 7,405 | 5.23M | BEIR |
| SciFact | 818 | 440 | 339 | 5K | BEIR |
| MS-Beir | 502,939 | 7,023 | 43 | 8.84M | BEIR |

Table 6: Statistics of the retrieval datasets used in our experiments.

### C.3.2 Details of MS-MARCO Subsets Construction

To efficiently evaluate DeepRetrieval on MS-MARCO (Nguyen et al., 2016), we created three topic-based subsets: Health (H), Science (S), and Technology (T). We merged training and development query splits and embedded them using Contriever (Izacard et al., 2022). We then applied K-means clustering (Hartigan & Wong, 1979) to create 25 clusters.

A human annotator randomly selected and labeled 10 clusters from the 25 total clusters with topic names. This was done by carefully examining the topic content of queries within each selected cluster. This annotation process allowed us to identify high-quality domain-specific queries for Health, Science, and Technology.

We maintained the original train-dev split mapping to prevent data leakage, resulting in:

- **MS-H**: 30,385 training and 2,034 development queries
- **MS-S**: 22,416 training and 2,305 development queries
- **MS-T**: 10,974 training and 1,291 development queries

We further constructed the corpus subsets to improve the efficiency of dense retrieval, we first included all groundtruth passages for the queries in the selected subsets, then randomly added noise passages from the entire corpus (with 8,841,822 passages in total) to reach a full collection of 800,000 passages.

### C.4 SQL Database Search

To evaluate the performance of DeepRetrieval in SQL database search, we use the Spider (Yu et al., 2018) and BIRD (Li et al., 2023) datasets.

Spider is a large-scale benchmark for text-to-SQL generation, consisting of 10,181 natural language questions paired with 5,693 unique, complex SQL queries spanning 200 databases across 138 different domains. Each database contains multiple tables, providing a diverse and challenging testbed for SQL generation.

BIRD is a more extensive dataset designed for SQL generation, containing 12,751 unique question-SQL pairs across 95 large databases, amounting to 33.4 GB of data. The dataset covers over 37 professional domains, such as blockchain, hockey, healthcare, and education. Unlike Spider, BIRD incorporates external knowledge to help models generate more precise SQL queries. As with Spider, the databases in these splits do not overlap. BIRD features more intricate SQL queries compared to Spider, making it a more challenging benchmark for SQL generation tasks.

The official splits of the two datasets are as follows:

| Dataset | # Train Examples | # Dev Examples | # Test Examples | # Database | # Table/DB | Source |
|---|---|---|---|---|---|---|
| Spider | 8,659 | 1,034 | 2,147 | 200 | 5.1 | Spider |
| BIRD | 9,428 | 1,534 | N/A | 95 | 7.3 | BIRD |

Table 7: Statistics of the BIRD and Spider datasets. "Table/DB" refers to the number of tables per database.

Notably, the databases in these splits are non-overlapping, ensuring a realistic evaluation of generalization. Following previous studies, for Spider, we use the training set to train all models and the test set for performance evaluation. For BIRD, we use the training set to train all models and the development set for performance evaluation.

In our experiments, we provide only the full database schema without using any actual database entries. The database schema describes the metadata of all tables in a database, including the column names and their data types. To compare RL with SFT, we further train supervised fine-tuning (SFT) models, which uses the ground-truth SQL queries from these datasets as well as the reasoning processes distilled from GPT-4o.

We show examples of each dataset below:

---

**Query:**
"Count the number of clubs."

**Database Schema:**
```
CREATE TABLE club (
    Club_ID               INTEGER not null primary key,
    Name                  TEXT,
    Manager               TEXT,
    Captain               TEXT,
    Manufacturer          TEXT,
    Sponsor               TEXT
);

CREATE TABLE player (
    Player_ID             REAL not null primary key,
    Name                  TEXT,
    Country               TEXT,
    Earnings              REAL,
    Events_number         INTEGER,
    Wins_count            INTEGER,
    Club_ID               INTEGER,
    FOREIGN KEY (Club_ID) REFERENCES club(Club_ID)
);
```
**Groundtruth SQL:**
SELECT count(*) FROM club;

**Groundtruth Data:**
9

---

Figure 10: A data example in Spider.

**Query:**
"What is the URL to the rating on Mubi made by user 45579900 for the movie 'The Vertical Ray of the Sun' that received 20 likes?"

**Database Schema:**

```
CREATE TABLE lists (
    ......
);
CREATE TABLE movies (
    movie_id                INTEGER not null primary key,
    movie_title             TEXT,
    movie_release_year      INTEGER,
    movie_url               TEXT,
    ......
);
CREATE TABLE ratings_users (
    ......
);
CREATE TABLE lists_users (
    ......
);
CREATE TABLE ratings (
    movie_id                INTEGER,
    rating_id               INTEGER,
    rating_url              TEXT,
    critic_likes            INTEGER,
    user_id                 INTEGER,
    ......
);
```

**Knowledge:**
"URL refer to rating_url; 20 likes refer to critic_likes = '20'; user 45579900 refer to user_id"

**Groundtruth SQL:**
SELECT T2.rating_url FROM movies AS T1 INNER JOIN ratings AS T2 ON T1.movie_id = T2.movie_id WHERE T2.user_id = 45579900 AND T1.movie_title = 'The Vertical Ray of the Sun' AND T2.critic_likes = 20;

**Groundtruth Data:**
[http://mubi.com/films/the-vertical-ray-of-the-sun/ratings/3580526]

Figure 11: A data example in BIRD.

# D  Additional Results and Analysis

## D.1  Literature Search

**Reasoning Is Helpful, Yet Its Length Shrinks Over Time.**  While our results clearly indicate that incorporating reasoning improves performance, we observe an intriguing trend: the length of the reasoning steps decreases over time. This is in contrast to the "aha moment" phenomenon observed in DeepSeek-R1 (Guo et al., 2025), where the reasoning chains became increasingly longer as training progressed. Instead, in our task, we find that the model's reasoning steps become more concise. A plausible explanation for this behavior lies in the nature of the task. Unlike tasks that demand explicit multi-step reasoning, such as math problems and coding, our retrieval-focused setup does **not inherently require long-form reasoning.** During early training (e.g., roughly Step 0 to 50, as shown in Figure 5(a)), the reasoning process plays a crucial role in exploration. It encourages the model to generate more diverse and semantically rich queries, helping it better understand the search engine's preferences. This leads to improved reward signals and better downstream performance. However, as training progresses and the model internalizes an effective retrieval strategy, it becomes increasingly capable of generating high-reward outputs with much shorter reasoning steps—or even skipping reasoning entirely for certain samples. Since the reward is primarily tied to the <answer> portion, the critic model gradually assigns lower value estimates to tokens in the <think> portion. This dynamic leads to a steady reduction in reasoning length (starting from around step 50) until it stabilizes.

**Without Reasoning, Model Falls into Local Minimum.**  In Figure 4(b) and 4(c), we observe that Qwen without reasoning increases its query length, which pushed it into a suboptimal trajectory. Thus, we conduct a study on this by sampling the most common rewarding patterns for all three models at step 400 and step 600, as shown in Figure 12.

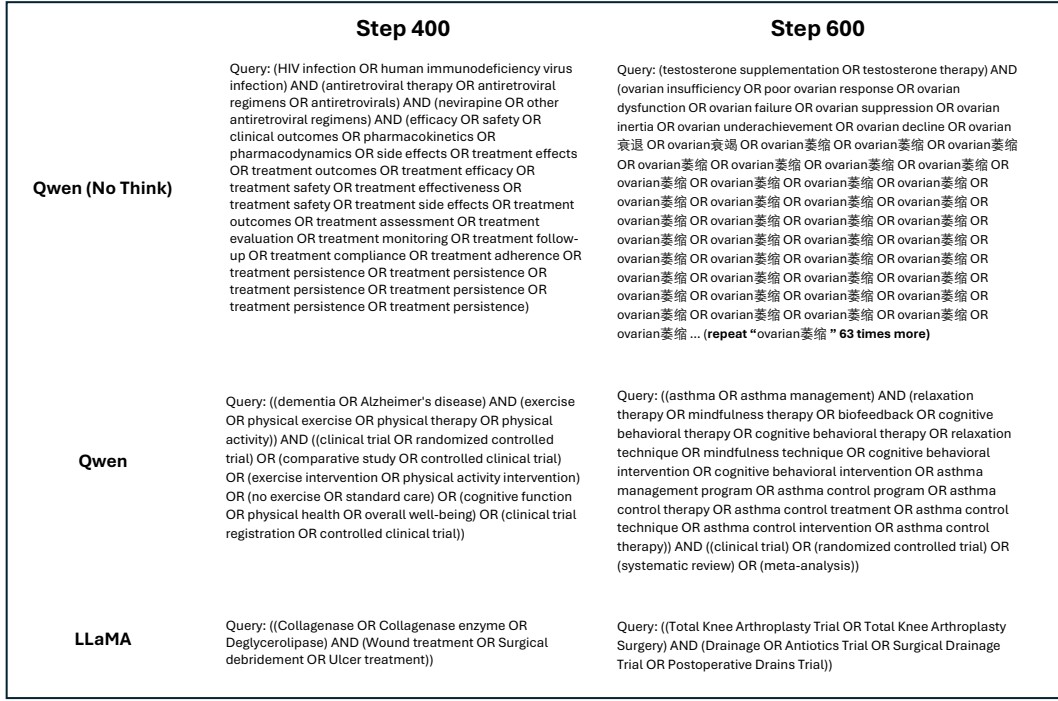

Figure 12: Query Generation Examples. We select the most common rewarded query patterns at step 400 and step 600 for all the models. "No Think" denotes no reasoning.

We observe that Qwen without reasoning was rewarded for long and verbose queries at the early stage (step 400), which causes the model to adopt an excessive expansion strategy. This

early reinforcement of verbosity leads the model to generate increasingly repetitive queries with redundant terms by step 600 (e.g., repeating "ovarian atrophy" numerous times). In contrast, Qwen and LLaMA with thinking capabilities produce high-quality, well-structured queries even at the early stage, which helps them continuously optimize toward more effective and semantically precise formulations. The reasoning process enables these models to explore the search space more systematically and focus on semantic relevance rather than mere term frequency, allowing them to avoid the degenerate strategy of query expansion through repetition. This demonstrates how incorporating reasoning from the beginning establishes a better optimization trajectory that leads to superior performance.

**Different LLMs Can Learn Distinct Policies with Similarly Strong Performance** In Figure 4(c), we observe that Qwen-2.5-3B-Instruct and Llama-3-3B reach comparable levels of performance between steps 1400 and 1800, both achieving strong retrieval results. However, when we examine Figure 4(b), we see a clear difference in their behavior: Qwen generates significantly longer rewritten queries, whereas Llama tends to produce shorter ones on average. This suggests that different LLMs can learn distinct policies through interactions with the same search engine—yet still arrive at similarly effective solutions. This highlights the flexibility of reinforcement learning in allowing models to discover multiple valid paths to high performance, depending on their internal representations and learning dynamics.

## D.2 Classic Sparse & Dense Text Retrieval

From Table 2, we also note that SciFact, unlike most other datasets, does not reach the best performance with DeepRetrieval in either sparse or dense settings. A likely explanation is that SciFact has a much smaller training set—only 800 examples—compared to thousands of training examples available in other datasets. This suggests that while DeepRetrieval can operate in low-resource settings compared with original BM25, it benefits more when a moderate amount of training data (e.g., a few thousand examples) is available. Investigating how the size and quality of training data impacts the effectiveness of DeepRetrieval is another promising avenue for future work.

## D.3 SQL Database Search

The task of SQL generation in BIRD presents greater challenges compared to Spider, resulting in overall lower performance for the same models. The effectiveness of RL is particularly sensitive to the interplay between a model's intrinsic capabilities and the complexity of the task. This is because RL training enables LLMs to explore strategies for maximizing rewards, and the exploration phase in the early stages of RL can significantly influence the entire training process. As shown in Table 3, Qwen2.5-Coder$_{3B-Inst}$, after RL training, surpasses GPT-4o (73.50) with an execution accuracy of 74.85. This highlights the effectiveness of RL in enhancing the SQL generation capabilities of the 3B base model. However, its performance on the BIRD dataset remains lower than that of GPT-4o, indicating that the additional complexity of BIRD poses greater challenges even for models trained with RL.

To address the challenges of RL training, cold start (Guo et al., 2025) has been introduced as a technique to guide the model toward more optimal exploration paths during the initial stages of RL. Cold start training extends a pre-SFT (Supervised Fine-Tuning) model, allowing it to explore beyond memorized patterns and discover novel strategies. From Table 3, we observe that the performance improvement from cold start is more pronounced in BIRD than in Spider. Specifically, for Qwen2.5$_{3B-Inst}$, cold start leads to a 6.3% increase in execution accuracy on BIRD (from 41.40 to 44.00) and a smaller 2.2% increase on Spider (from 68.79 to 70.33). A similar trend is observed for Qwen2.5-Coder$_{3B-Inst}$.

Furthermore, we find that knowledge-rich base models exhibit less reliance on cold start. For example, while DeepRetrieval$_{3B}$ benefits from cold start, DeepRetrieval$_{3B-Coder}$ shows only marginal improvement on BIRD (from 49.02 to 50.52) and even experiences a slight performance drop on Spider (from 74.85 to 74.34). This suggests that models with strong prior coding capabilities can effectively explore optimal SQL generation strategies without requiring additional supervision during the cold start phase.

Our ablation study further investigates the role of model reasoning in SQL generation during RL training. While reasoning generally enhances performance, its absence leads to only minor degradation overall. Notably, DeepRetrieval$_{3B\text{-}Base}$ exhibits an improvement in execution accuracy (from 68.79% to 70.24%) when reasoning is removed, aligning with the findings of Wu et al. (2025), which suggest that shorter reasoning chains may be beneficial for simpler tasks. Conversely, DeepRetrieval$_{3B\text{-}Coder}$, which is pre-trained on complex code generation, leverages its inherent coding abilities and benefits from reasoning during RL. These results indicate that for relatively straightforward SQL generation tasks, such as those in Spider, reasoning during RL may not be essential, particularly for models lacking strong coding capabilities.

# E   Knowledge Injection Analysis

We define *knowledge injection* as the phenomenon where LLMs incorporate answer information into generated queries based on their prior knowledge, potentially artificially inflating retrieval performance. This appendix describes our method for quantifying and analyzing this effect.

## E.1   Analysis Methodology

We developed a post-processing pipeline to detect and analyze knowledge injection:

1. We use Claude-3.5-Sonnet as an auxiliary LLM to analyze each generated query, determining if it contains spans that directly match answer candidates and cannot be derived from the original query without prior knowledge.
2. For queries containing injected knowledge, we extract a "cleaned" version with the injected spans removed.
3. We evaluate both original and cleaned queries, comparing their performance to measure the impact of knowledge injection.

## E.2   Implementation

The detection process uses a specialized prompt (shown in Figure 18) that instructs the auxiliary LLM (Claude-3.5-Sonnet) to analyze queries and provide:

- Whether the query contains answer spans
- Whether these spans could be derived from the original query
- A list of identified answer spans
- A cleaned version of the query

We implemented this analysis using a batch processing framework that efficiently handles large numbers of queries in parallel. For each model and dataset combination, we computed:

- **Injection Rate**: Percentage of queries containing injected knowledge
- **Performance Delta**: Difference in retrieval accuracy between original and cleaned queries

## E.3   Key Findings

Our analysis revealed distinct patterns of knowledge injection across datasets:

- **Natural Questions**: Performance gains came from both knowledge injection (33.4% of queries) and other factors such as query reformulation. Performance remained relatively strong even after removing injected content.
- **TriviaQA**: Improvements were heavily dependent on knowledge injection (56.9% of queries), with significant performance drops when injected content was removed.

- **SQuAD**: Despite high injection rates from some models, knowledge injection had minimal impact on performance. DeepRetrieval showed minimal injection (10.5%), suggesting success on this dataset relies more on understanding dataset distribution than prior knowledge.

These findings demonstrate that DeepRetrieval effectively adapts its augmentation strategy to each dataset's characteristics, explaining its consistent performance advantages across diverse retrieval tasks.

The results also highlight the importance of considering knowledge injection when evaluating LLM-based query augmentation systems, as raw performance metrics alone may not provide a complete picture of model capabilities.

## F  Task Definitions, Metrics, and Retrieval Rewards

| Task | Definition | Evaluation Metric | Retrieval Reward | |
|------|-----------|-------------------|-------------------|---|
| Literature Search | Given a query, retrieve relevant documents using search engine | Recall@K (percentage of groundtruth documents retrieved in top K results) | 5.0
4.0
3.0
1.0
0.5
0.1
-3.5 | if recall $\geq$ 0.7
if recall $\geq$ 0.5
if recall $\geq$ 0.4
if recall $\geq$ 0.3
if recall $\geq$ 0.1
if recall $\geq$ 0.05
otherwise |
| Evidence-Seeking | Given a query (question), retrieve documents containing answer spans that match any answer candidates | H@N (whether the rank of the first occurred answer span is before N) | 5.0
4.0
2.0
1.0
0.5
0.1
-3.5 | if rank $\leq$ 5
if rank $\leq$ 20
if rank $\leq$ 50
if rank $\leq$ 100
if rank $\leq$ 1000
if rank $\leq$ 3000
otherwise |
| Sparse Retrieval | Given a query, retrieve relevant documents using keyword matching and boolean operations with BM25 | NDCG (Normalized Discounted Cumulative Gain; measures ranking quality with relevance grades) | Same as evaluation metric (NDCG) | |
| Dense Retrieval | Given a query, retrieve relevant documents using semantic vector representations of queries and documents | NDCG (Normalized Discounted Cumulative Gain; prioritizes relevant documents appearing higher in results) | Same as evaluation metric (NDCG) | |
| Database Search | Given a natural language query, generate SQL to search a database for the answer | Execution accuracy (match between retrieved and groundtruth answers) | Same as evaluation metric (execution accuracy) | |

Table 8: Task definitions, evaluation metrics, and reward designs for different retrieval tasks.

In Table 8, we list task definitions, their evaluation metrics, and the retrieval rewards we designed for them.

In the task of SQL database search on the BIRD dataset, we found that the initial stage of RL for the models did not perform well. Due to the increased difficulty of the BIRD dataset, LLMs may struggle to effectively explore the optimal search space in the beginning. To address this, we added the successful execution of the generated SQL as an additional reward, with a value of 0.3 for these models, to enhance their RL training process. This reward is granted when the generated SQL contains no syntax errors or missing tables.

# G Query Generation and Retrieval Examples

## G.1 Literature Search

| Generator | Case |
|---|---|
| - | **Original Query:** 
 "P: Patients undergoing perioperative procedures, I: Desmopressin administration, C: Standard care without desmopressin, O: Minimising perioperative allogeneic blood transfusion" 

 **Targets Document PubMed IDs:** [3286039, 11704449, 1863725, 1529684, 1622045, 2521029, 1610009, 11698939, 2682243, 1472662, 8937605, 8400098, 8306217, 1903723, 7475139, 8202769, 10408485, 7519132, 1434725, 3517650, 14980901, 1934382, 2405964, 1614196, 2042789, 7674464, 7572006] |
| - | **"OR"-Connected Query:** 
 "Patients undergoing perioperative procedures OR Desmopressin administration OR Standard care without desmopressin OR Minimising perioperative allogeneic blood transfusion" 

 **Document Hits:** [3286039, 11704449, 1863725, 1529684, 1622045, 2521029, 1610009, 11698939, 2682243, 1472662, 8937605, 8400098, 8306217, 1903723, 7475139, 8202769, 10408485, 7519132, 1434725, 3517650, 14980901, 1934382, 2405964, 1614196, 2042789, 7674464, 7572006] 

 **Recall@3K:** 0.0% |
| GPT-4o | **Generated Query:** 
 "(perioperative AND desmopressin AND blood transfusion)" 

 **Document Hits:** [3286039, 11704449, **1863725**, 1529684, 1622045, 2521029, 1610009, 11698939, 2682243, 1472662, **8937605**, **8400098**, 8306217, 1903723, 7475139, 8202769, 10408485, 7519132, 1434725, 3517650, 14980901, 1934382, **2405964**, 1614196, 2042789, 7674464, 7572006] 

 **Recall@3K:** 14.81% |
| Sonnet-3.5 | **Generated Query:** 
 "(perioperative OR surgery) AND (desmopressin OR DDAVP) AND (blood transfusion)" 

 **Document Hits:** [**3286039**, **11704449**, **1863725**, 1529684, **1622045**, 2521029, **1610009**, **11698939**, **2682243**, 1472662, **8937605**, **8400098**, **8306217**, **1903723**, **7475139**, **8202769**, **10408485**, **7519132**, 1434725, 3517650, **14980901**, **1934382**, **2405964**, **1614196**, 2042789, **7674464**, 7572006] 

 **Recall@3K:** 74.07% |
| DeepRetrieval | **Generated Query:** 
 "((DDAVP) AND (Perioperative Procedures OR Blood Transfusion OR Desmopressin OR Anticoagulant)) AND (Randomized Controlled Trial)" 

 **Document Hits:** [**3286039**, **11704449**, 1863725, **1529684**, **1622045**, **2521029**, **1610009**, **11698939**, **2682243**, **1472662**, **8937605**, **8400098**, **8306217**, **1903723**, **7475139**, **8202769**, **10408485**, **7519132**, **1434725**, **3517650**, **14980901**, **1934382**, **2405964**, **1614196**, **2042789**, **7674464**, **7572006**] 

 **Recall@3K:** 96.30% |

Table 9: Case Study of Literature Searching on PubMed Search Engine.

## G.2 Evidence-Seeking Retrieval

| Generator | Case |
|---|---|
| - | **Original Query:** "What is another term for the pivot mounting?" 

 **Answer Candidates:** ["trunnion"] 

 **First occurred document containing an answer candidate:** 
 "Hydraulic cylinders with a long stroke and large bore can be unstable. Side mounts need to be well aligned, with the load properly supported and guided. Centerline lug mounts absorb forces along the centerline and require dowel pins to secure the lugs, preventing movement when operating at higher pressures or under shock conditions. Pivot mounts absorb force on the cylinder centerline and allow the cylinder to change alignment in one plane. Common types include clevises, **trunnion** mounts, and spherical bearings. Since these mounts allow a cylinder to pivot, they should be used with rod-end connections." 

 **Rank:** 28 |
| GPT-4o | **Generated Query:** "pivot mounting OR pivot fastening OR pivot joint OR swivel mounting OR rotary mount OR hinge pivot" 

 **Answer Candidates:** ["trunnion"] 

 **First occurred document containing an answer candidate:** 
 "Hydraulic cylinders with a long stroke and large bore can be unstable. Side mounts need to be well aligned, with the load properly supported and guided. Centerline lug mounts absorb forces along the centerline and require dowel pins to secure the lugs, preventing movement when operating at higher pressures or under shock conditions. Pivot mounts absorb force on the cylinder centerline and allow the cylinder to change alignment in one plane. Common types include clevises, **trunnion** mounts, and spherical bearings. Since these mounts allow a cylinder to pivot, they should be used with rod-end connections." 

 **Rank:** 29 |
| DeepRetrieval | **Generated Query:** "(The alternative term for the pivot mounting, which is the pivot mounting)" 

 **Answer Candidates:** ["trunnion"] 

 **First occurred document containing an answer candidate:** 
 "Hydraulic cylinders with a long stroke and large bore can be unstable. Side mounts need to be well aligned, with the load properly supported and guided. Centerline lug mounts absorb forces along the centerline and require dowel pins to secure the lugs, preventing movement when operating at higher pressures or under shock conditions. Pivot mounts absorb force on the cylinder centerline and allow the cylinder to change alignment in one plane. Common types include clevises, **trunnion** mounts, and spherical bearings. Since these mounts allow a cylinder to pivot, they should be used with rod-end connections." 

 **Rank:** 17 |

Table 10: Case Study of Evidence-seeking Retrieval.

### G.3 Classic Sparse & Dense Document Retrieval

| Generator | Case |
|---|---|
| - | **Original Query:** "Ron Dennis is unemployed."

**Targets Document IDs:** [Ron Dennis]
**Retrieved Top-10 Document IDs:** ["Me, You and Him", "South African Unemployed Peoples' Movement", "Insider-outsider theory of employment", "Izuna: Legend of the Unemployed Ninja", "Quaesitor", "Community Action Programme", "National Unemployed Workers' Movement", "List of OECD countries by long-term unemployment rate", "Jack Murphy (Irish politician)", "Active labour market policies"]
**Document** #{"Me, You and Him"}: "Me, You and Him Me. You and Him is a British television sitcom , that aired on ITV from 30 July to 3 September 1992... Punt 's character , the **unemployed** but intelligent Mark Prior , lives nearby but was forever visiting the others after arguing with his parents..."

**NDCG@10:** 0.00 |
| GPT-4o | **Generated Query:** "("Ron Dennis" AND ("unemployed" OR "jobless" OR "without job" OR "out of work" OR "not employed" OR "without employment" OR "laid off" OR "seeking employment" OR "looking for work"))"

**Targets Document IDs:** [Ron Dennis]
**Retrieved Top-10 Document IDs:** ["Job guarantee", "Youth unemployment", "Work permit (United Kingdom)", "Informational interview", "Unemployment in Ontario", "Job-seeking expense tax deductions", "United States Employment Service", "Workfare", "Active labour market policies", "Job hunting" ]
**Document** #{"Job guarantee"}: "Job guarantee. A job guarantee (JG) is an economic policy proposal aimed at providing a sustainable solution to the dual problems of inflation and **unemployment**... In Marxian terms, the unemployed serve as a reserve army of labor. By contrast, in a job guarantee program, a buffer stock of employed people ( employed in the job guarantee program ) provides the same protection against inflation without the social costs of **unemployment**, hence potentially fulfilling the dual mandate of full employment and price stability."

**NDCG@10:** 0.00 |
| DeepRetrieval | **Generated Query:** "((Ron Dennis IS-A person AND Ron Dennis IS-A employed) OR (Ron Dennis IS-A person AND Ron Dennis IS-A unemployed)))"

**Targets Document IDs:** [Ron Dennis]
**Retrieved Top-10 Document IDs:** ["Ron Dennis", "McLaren Technology Group", "Dennis A. Murphy Trophy", "Dennis Leeflang", "Cutaway (2000 film)", "Dennis Hextall", "WKAN", "Rondel Racing", "New Race", "Hextall" ]
**Document** #{"Ron Dennis"}: "Ron Dennis. Ronald Dennis CBE ( born 1 June 1947 ) is a British businessman and Official British Business Ambassador for The United Kingdom. Dennis is the Global Consultant for Minsheng Investment Corporation and also owner of Absolute Taste. He is best known for his former role as owner, CEO, chairman and founder of McLaren Technology Group. Dennis was removed from his McLaren management roles in 2016 but remains a director of the company and a 25% shareholder. Between 1981 and 2009, Dennis was the team principal of the McLaren Formula One team and was instrumental in transforming the outfit into a regular world championship contender. Constructors ' and drivers ' world championships were won with Niki Lauda , Alain Prost , Ayrton Senna , Mika Häkkinen and Lewis Hamilton ."

**NDCG@10:** 1.00 |

Table 11: Case Study of Classic Sparse Retrieval on FEVER dataset.

| Generator | Case |
|---|---|
| - | **Original Query:** "Who is older, Annie Morton or Terry Richardson?"

**Targets Document IDs:** [ "39354179", "1316127"]
**Retrieved Top-10 Document IDs:** [ **"1316127"**, "31582340", "43900", **"39354179"**, "23960843", "53800293", "26314230", "12795477", "3723961", "443375"]
**Document** #{"1316127"}: "Terry RichardsonTerrence "Uncle Terry" Richardson (born August 14, 1965) is an American fashion and portrait photographer who has shot advertising campaigns for Marc Jacobs, Aldo, Supreme, Sisley, Tom Ford, and Yves Saint Laurent among others. He has also done work for magazines such as ..."

**NDCG@10:** 0.8772 |
| GPT-4o | **Generated Query:** "Annie Morton birth date or age, Terry Richardson birth date or age, compare ages, who is older"

**Targets Document IDs:** [ "39354179", "1316127"]
**Retrieved Top-10 Document IDs:** ["6271033", "23960843", "29475444", "26314230", "8190335", "53402227", "53235326", "594667", "23121469", "9389734"]
Document #{"6271033"}: Annie KnightAnnie Knight (6 June 1895 – 27 November 2006) was, at age 111 years 174 days, the United Kingdom's oldest person following the death of fellow 111-year-old Emmeline Brice on 26 July 2006.

**NDCG@10:** 0.00 |
| DeepRetrieval | **Generated Query:** "Who is older, Annie Morton or Terry Richardson?"

**Targets Document IDs:** ["39354179", "1316127"]
**Retrieved Top-10 Document IDs:** [**"1316127"**, "31582340", "43900", **"39354179"**, "23960843", "53800293", "26314230", "12795477", "3723961", "443375"]
**Document** #{"1316127"}: "Terry RichardsonTerrence "Uncle Terry" Richardson (born August 14, 1965) is an American fashion and portrait photographer who has shot advertising campaigns for Marc Jacobs, Aldo, Supreme, Sisley, Tom Ford, and Yves Saint Laurent among others. He has also done work for magazines such as ..."

**NDCG@10:** 0.8772 |

Table 12: Case Study of Classic Dense Retrieval on HotpotQA.

## G.4 SQL Database Search

| Generator | Case |
|---|---|
| - | **Original Query:** "What are the titles of books that are not "Poet"?" 

 **Database Schema:** 

 ```PRAGMA foreign_keys = ON;```

```CREATE TABLE "book" (```
```  "Book_ID" int,```
```  "Title" text,```
```  "Type" text,```
```  "Pages" int,```
```  "Chapters" int,```
```  "Audio" text,```
```  "Release" text,```
```  PRIMARY KEY ("Book_ID")```
```);```

```CREATE TABLE "review" (```
```  "Review_ID" int,```
```  "Book_ID" int,```
```  "Rating" real,```
```  "Readers_in_Million" real,```
```  "Rank" int,```
```  PRIMARY KEY ("Review_ID"),```
```  FOREIGN KEY ("Book_ID") REFERENCES `book`("Book_ID")```
```);```

 **Groundtruth SQL:** SELECT Title FROM book WHERE TYPE != 'Poet'; 
 **Targets Data:** ['A Game of Thrones', 'A Clash of Kings', 'A Storm of Swords', 'A Feast for Crows'] |
| GPT-4o | **Think:** "To solve the query, we need to retrieve the titles of books from the "book" table where the "Title" is not equal to "Poet". We will use a SELECT statement to get the "Title" column and a WHERE clause to filter out the books where the "Title" is "Poet". The condition for the WHERE clause will be "Title != 'Poet'". The SQL query should be straightforward since we are only working with one table and one condition." 
 **Generated SQL Query:** SELECT Title FROM book WHERE Title != 'Poet'; 

 **Retrieved data:** [] (empty) 

 **Execution Accuracy:** 0.0 |
| DeepRetrieval | **Think:** "To achieve this, we need to select the Title column from the book table. Then, we add a WHERE clause to filter out the books where the Type column does not match the value 'Poet'." 
 **Generated SQL Query:** SELECT Title FROM book WHERE Type != 'Poet'; 

 **Retrieved data:** ['A Game of Thrones', 'A Clash of Kings', 'A Storm of Swords', 'A Feast for Crows'] 

 **Execution Accuracy:** 1.0 |

Table 13: Case Study of SQL Database Search.

# H   Prompts

In this section, we showcase the prompts we used in this study.

**Literature Searching**

A conversation between User and Assistant. The User asks a question, and the Assistant solves it. The Assistant first thinks about the reasoning process in the mind and then provides the User with the answer. The Assistant is a clinical specialist. He is conducting research and doing a medical literature review. His task is to create query terms for a search URL to find relevant literature on PubMed or ClinicalTrials.gov.

The research is defined using the PICO framework:
P: Patient, Problem or Population - Who or what is the research about?
I: Intervention - What is the main intervention or exposure being considered?
C: Comparison - What is the intervention being compared to?
O: Outcome - What are the relevant outcomes or effects being measured?

The Assistant should show his thinking process in <think> </think> tags. The Assistant should return the final answer in JSON format in <answer> </answer> tags,
For example:
<think>
[thinking process]
</think>
<answer>
{
"query": "...."
}
}
</answer>.
Note: The query should use Boolean operators (AND, OR) and parentheses for grouping terms appropriately.

User: The research is defined by the following PICO:

P: {P}
I: {I}
C: {C}
O: {O}

Assistant: Let me solve this step by step.
<think>

**Example**

P: *Men with subfertility due to oxidative stress*
I: *Oral supplementation with antioxidants*
C: *No antioxidant supplementation or placebo*
O: *Improvement in sperm quality by reducing oxidative damage*

Figure 13: Prompt used for literature search.

**Evidence-Seeking Retrieval**

You are a helpful assistant. You first thinks about the reasoning process in the mind and then provides the user with the answer.
You are a query rewriting expert. Your task is to create query terms for user query to find relevant literature in a Wikipedia corpus using BM25.

Show your work in <think> </think> tags. Your final response must be in JSON format within <answer> </answer> tags. For example,
<think>
[thinking process]
</think>
<answer>
{
"query": "...."
}
</answer>.
Note: The query should use Boolean operators (AND, OR) and parentheses for grouping terms appropriately.

Here's the user query:
{User Query}

Example (NQ)

*what's the dog's name on tom and jerry*

Assistant: Let me rewrite the query with reasoning.
<think>

Figure 14: Prompt used for evidence-seeking retrieval.

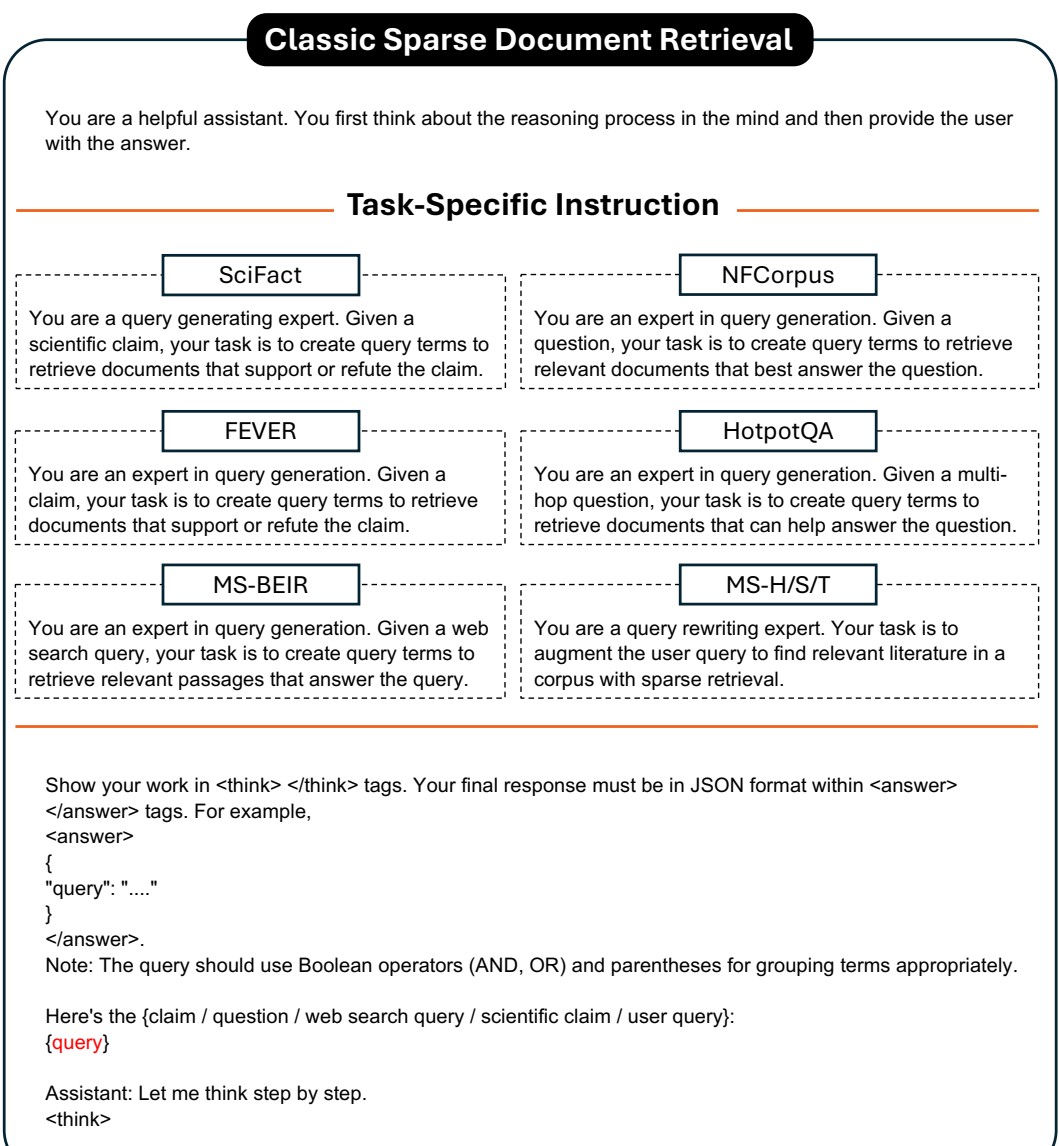

Figure 15: Prompt used for classic sparse document retrieval.

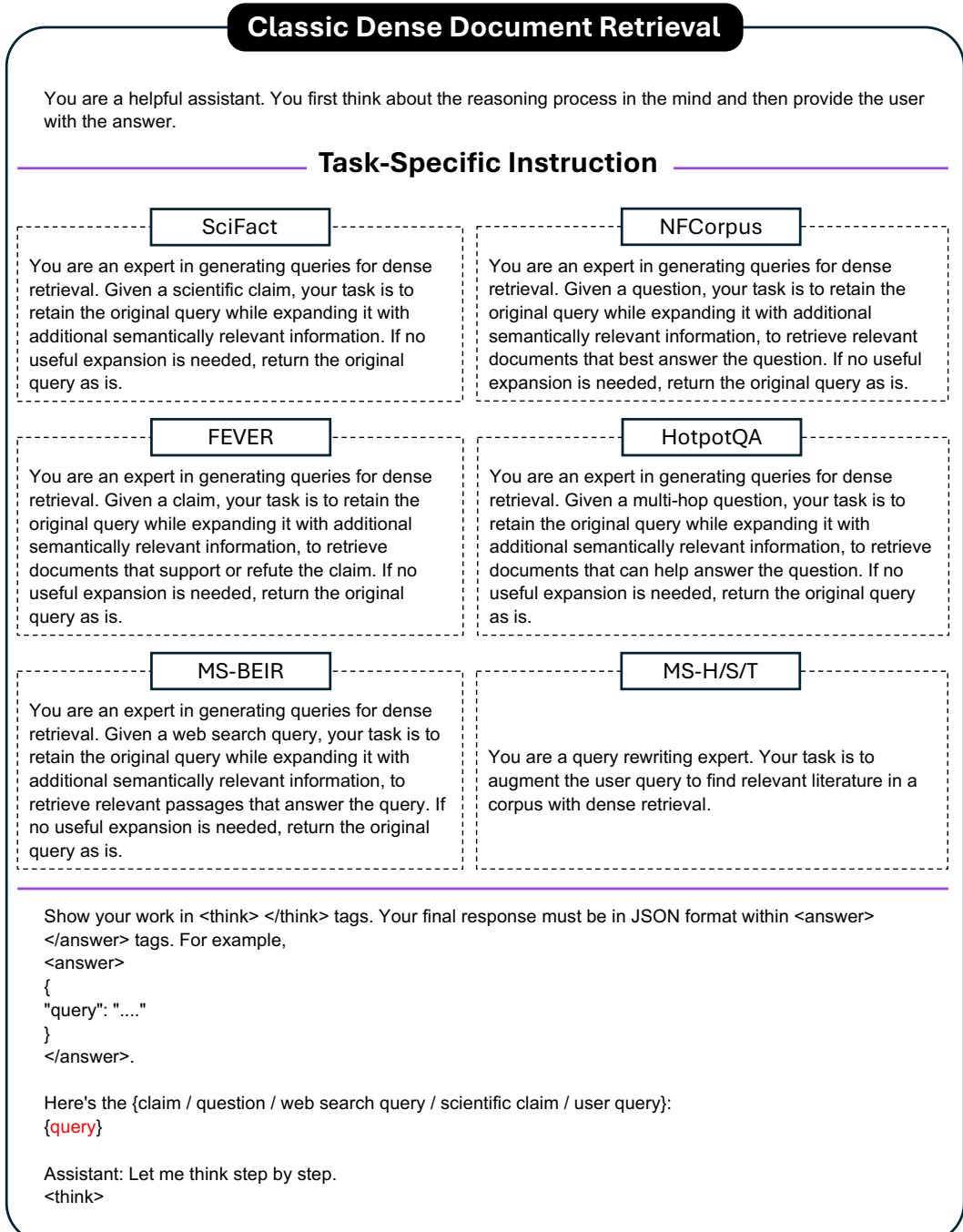

Figure 16: Prompt used for classic dense document retrieval.

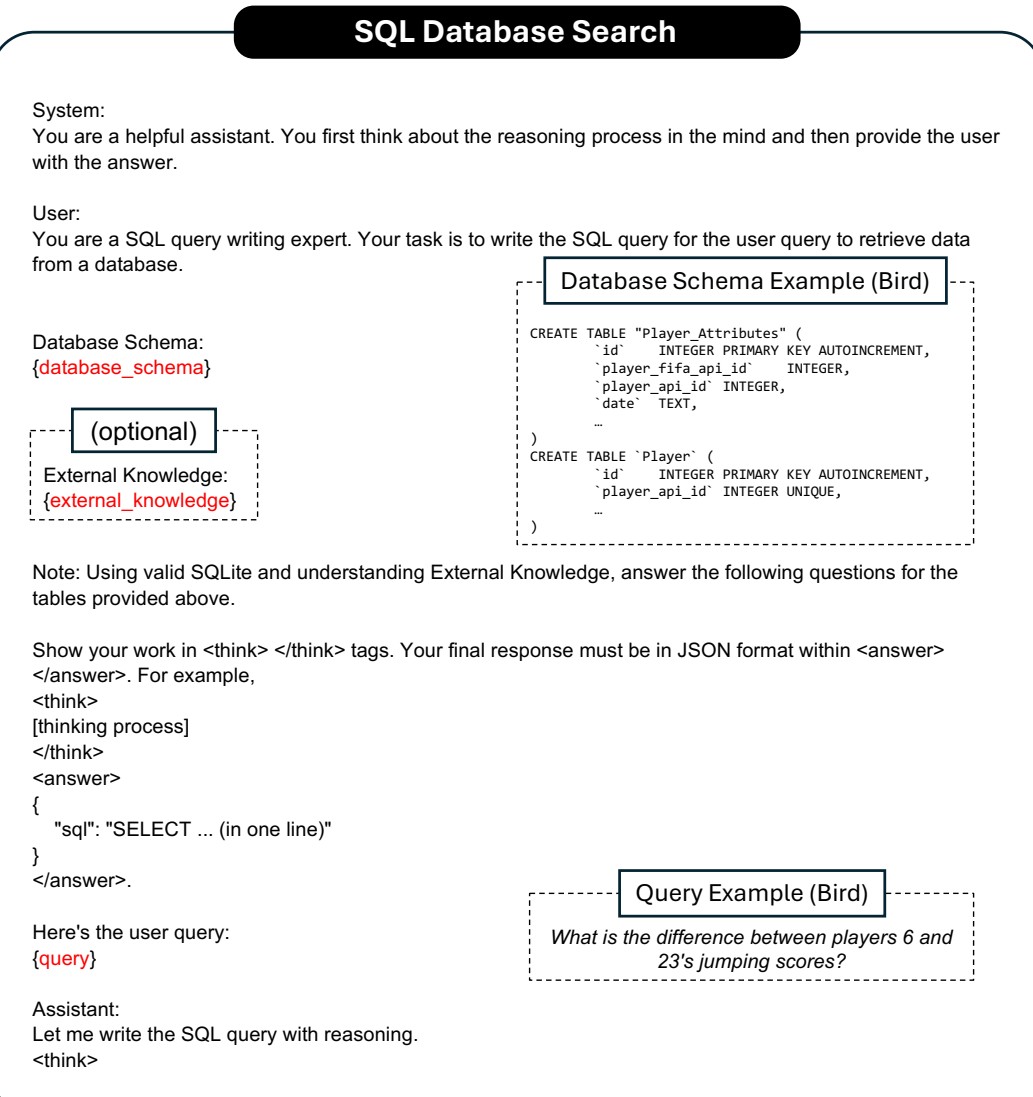

Figure 17: Prompt used for SQL database search.

---

**Knowledge Injection Check & Clean**

You are a helpful assistant that checks the quality of query augmentation. As we use LLM to augment the query, we need to check if the augmented query can be derived from the original query.
Your task is to analyze if there are answer spans in the query that directly match the answer candidates and cannot be derived from the original query without using prior knowledge.

Instructions:
1. Check if any part of the augmented query exactly matches or paraphrases any answer candidate, if so, set "has_answer" to true.
2. If the augmented query cannot be derived from the original query without using prior knowledge, set "cannot_be_derived" to true.
3. If both "has_answer" and "cannot_be_derived" are true, remove those answer spans from the query, without changing other parts of the augmented query.
4. Return your analysis in a strict JSON format

IMPORTANT:
You must respond with valid JSON wrapped in <answer> tags. The JSON must have this exact structure:

The content between <answer> and </answer> MUST be a valid JSON object:
- Use double quotes for strings
- Escape special characters with backslashes (e.g., \\" for quotes within strings)
- Follow standard JSON formatting rules

Example valid response:
For the following original query:
"How many people live in New York City in 2020?"

and the augmented query:
("population" OR "residents") AND "new york city" AND "2020" AND "8 million"

Answer candidates to check against:
["8.253 million", "8,253,213", "8.25M", "approximately 8.25 million"]

<answer>
{
"has_answer": true,
"cannot_be_derived": true,
"answer_span_in_query": ["8 million"],
"matched_answer_candidates": ["approximately 8.25 million", "8.25M"],
"cleaned_query": "(\"population\" OR \"residents\") AND \"new york city\" AND \"2020\""
}
</answer>

Now, your turn:

Original query:
{original_query}
Augmented query to analyze:
{augmented_query}

Answer candidates to check against:
{answer_candidates}

Your response:

Figure 18: Prompt used for knowledge injection check and clean.

## I   Contribution Statement

**P. Jiang** developed the DeepRetrieval framework and conducted the preliminary study on literature search using search engines. He was responsible for writing the abstract, introduction, methodology, main results, and discussion sections. He carried out all experiments on literature search (in Table 1), evidence-seeking retrieval (in Table 1), and sparse retrieval across the MS-H, MS-S, and MS-T datasets (in Table 2). Additionally, he conducted the knowledge injection (Figure 3) and think/query length studies/analyses (Figure 4), and created all figures presented in the paper.

**J. Lin** provided guidance on the VERL framework (the base architecture for the RL approach) and conducted the majority of the sparse and dense retrieval experiments (84 out of 96 in Table 2) following P. Jiang's preliminary findings on search engine retrieval. He drafted the complete methodology section and wrote the analysis on sparse and dense retrieval in the main results and discussion sections, as well as analysis sections in the Appendix (the first analysis in D.1 and D.2).

**L. Cao** implemented the SQL database search component and conducted all experiments on the BIRD and Spider datasets (in Table 3). He also constructed the MS-H, MS-S, and MS-T subsets and performed dense retrieval experiments (in Table 2) on these datasets. Additionally, he contributed to the main results, discussion, and analysis sections related to SQL retrieval.

**R. Tian** participated in project discussions, conducted the literature review, and authored the majority content of the Related Work section (Appendix §A).

**S. Kang** joined the project after the preliminary results on search engine retrieval, provided suggestions based on retrieval literature, and proposed datasets for evaluating the method's performance. He also suggested relevant papers for discussion in the Related Work section, and contributed to paper editing.

**Z. Wang** initiated the idea of applying reinforcement learning to the literature search task and contributed to project discussions. He also proposed the performance visualization format and assisted in paper editing.

**Prof. J. Sun** and **Prof. J. Han** provided guidance on framework design, offered suggestions on paper writing, and contributed to paper editing.

