# OpenReview forum: "DeepRetrieval: Hacking Real Search Engines and Retrievers with Large Language Models via Reinforcement Learning"
_colmweb.org/COLM/2025/Conference — COLM 2025_

### Official Review · Reviewer_C5L2 · 2025-05-10

**Rating:** 7
**Confidence:** 4
**Ethics Flag:** 1

**Summary:**

The paper presents an ‘RL from scratch’ approach for reformulating information-seeking queries for retrieval tasks. Following a similar strategy used in reasoning models, e.g. DeepSeek-R1, the method samples several continuations, each instructed first to generate the thinking steps, followed by the updated query. Then, for each sample, a reward is calculated where the reward function is either recall@K or nDCG@k, depending on the task, with an additional format-based term to encourage the model to follow the given output structure. To compute reward, a retriever or search engine is used to return results based on rewritten queries. The rewards are subsequently computed with respect to the ground-truth.

The evaluation is done on literature search (where recall is important), information-seeking QA (where precision is important) and text-to-SQL. The results demonstrate the effectiveness of the proposed method using Qwen2.5-3B and Llama-3.2-3B models compared to strong proprietary LLMs (e.g. GPT-4o and Claude-3.5-Sonnet).

**Questions To Authors:**

- __Unclear retrieval in information-seeking QA:__  I could not find what retrieval model was used for the Evidence-Seeking tasks. Specifically, what retrieval models were used for the results in Table 1? Is it BM25 or a dense model?

- __How were the reformulated queries passed to the retriever model or the search engine?__ If I understood correctly, the paper does NOT do query expansion to append the resulting queries to the original ones before passing to the search engine. I think the authors can clarify how exactly the rewritten queries are used during inference time.

**Reasons To Accept:**

- __Interesting idea with convincing results:__ The use of RL in IR tasks is compelling and the strong empirical results show its effectiveness. This makes the work highly relevant and interesting to the community.

- __Comprehensive experiments:__ The experiments and the details provided in the appendix (despite some missing details) make the work reproducible, thus enabling the community to easily build upon this work.

**Reasons To Reject:**

- __Misleading claim about not using supervised data:__ While it is true that to update model parameters in an RL optimization step, the model does not require ground-truth labels, they are still needed for computing the reward (which is basically recall or nDCG in this case). Therefore, I find claims such as “without supervised data for reference query” in the paper misleading as they overlook the fact that supervised data is very much necessary for the approach to compute rewards.

- __Lack of discussion on training runtime:__ As intriguing as using RL in retrieval systems sounds, the reward computation is overwhelmingly slow due to the need for running retrieval (or calling APIs), which significantly slows down training. The paper  would be strengthened by a discussion on how long training takes and how many optimization steps are needed to achieve the reported results.

- __Use of outdated QA in evaluation:__ One main issue with using outdated datasets (NQ, TriviaQA, and SQuAD) is that many recent LLMs are directly or indirectly trained on them, enabling the LLMs to answer questions from these datasets entirely from their prior knowledge. This was also highlighted by the authors in the knowledge injection experiment (Figure 3). Another issue (though less prevalent) is that the answer to some of the questions in these datasets may have changed over time and the LLMs’ answers are actually correct but do not match the provided gold answers. Another issue with SQuAD is that it includes many questions whose answers are context-dependent, due to the nature of the reading comprehension task  (e.g. _What answer denotes that an algorithm has accepted an input string?_) which makes it less suitable for retrieval tasks.

---

> ### Author Response · Authors · 2025-05-30
> **[1/3] Addressing W1 - Clarifying “No Supervised Data”: No Reference Queries Used for Generation**
>
> **Dear Reviewer C5L2,**
>
> Thank you very much for your thoughtful and constructive review. We sincerely appreciate your recognition of our core contributions, particularly the novelty of the RL-based formulation, the strength of the results, and the breadth of our experimental study.
>
> Below we address each of your concerns in turn.
>
> ---
>
> > **[W1]** **Misleading claim about not using supervised data.**
>
> You're absolutely right that our phrasing could be clearer. In our paper, when we refer to “not using supervised data,” we specifically mean that we **do not use reference queries** for the query generation task $q \rightarrow q'$ where $q$ is the original query and $q'$ is the generated query.
>
> Prior works (see Appendix A.1 & A.3) often assume the existence of a gold-standard rewritten query $q^{\text{ref}}$, and optimize generation by minimizing a loss like:
> $\mathcal{L} = \text{CrossEntropy}(q', q^{\text{ref}})$
>
> **DeepRetrieval avoids this**: it does not require or assume any gold rewriting targets. Instead, the model learns through exploration — generating candidate queries $q'$ and receiving **task-specific reward signals** based on retrieval performance like $r(q,q') = \text{Recall}(q') \quad \text{or}\quad \text{NDCG}(q')$
>
> This reward is computed using ground-truth labels (e.g., relevant documents or answer spans), **but not any reference rewriting**. While this still involves some form of supervision, we emphasize that:
>
> > **These ground-truths are naturally available in most retrieval datasets and do not require any additional annotation effort.**
>
> In contrast, constructing reference queries requires expensive human labeling or distillation from large proprietary models (e.g., GPT-4o), which can be both costly and biased.
>
> That said, we agree that the phrase "without supervised data" could be misinterpreted without context. We’ll revise this in the final version to **“without supervised query rewriting pairs”** or **“without training on reference queries”** to be more precise!

---

> > ### Author Response · Authors · 2025-05-30
> > **[2/3] Addressing W2 – Training Efficiency: Runtime Decomposition and Practical Feasibility**
> >
> > (continue)
> >
> > > **[W2] Lack of discussion on training runtime.** (1) the reward computation is overwhelmingly slow due to the need for running retrieval (or calling APIs), which significantly slows down training. (2) The paper would be strengthened by a discussion on how long training takes and how many optimization steps are needed to achieve the reported results.
> >
> > ---
> >
> > **(1) Reward Computation Overhead**
> >
> > Thanks for flagging this. To demystify training efficiency, we now include a **runtime decomposition per optimization step** (batch size 64) across several retrieval tasks (**see full results in [b]**):
> >
> > | Task                        | Query Generation | Environment Interaction           | Parameter Update | Time  per Step (sec) |
> > | --------------------------- | ---------------- | -------------------------- | ---------------- | -------------------- |
> > | HotpotQA (IR-BM25)          | 9.1              | 10.0 (BM25 Retrieval)      | 29.4             | 57.9                 |
> > | HotpotQA (IR-Dense)         | 7.6              | 56.1 (Dense Retrieval)     | 25.3             | 94.6                 |
> > | PubMed (Literature Search)  | 11.5             | 104.4 (Search API calling) | 19.7             | 150.3                |
> > | TriviaQA (Evidence-Seeking) | 6.8              | 21.7 (BM25 Retrieval)                | 25.8             | 63.2                 |
> > | BIRD (SQL Search 3B)        | 14.6             | 33.2 (SQL Fetch)           | 37.0             | 108.2                |
> > | BIRD (SQL Search 7B)        | 24.5             | 35.2 (SQL Fetch)           | 82.1             | 173.4                |
> >
> > While PubMed search has the highest per-step time due to reward computation (104.4 sec out of 150.3), this is mainly caused by **rate limits of the public PubMed API**. In enterprise settings with higher-rate access, **retrieval speed can scale linearly**. For other tasks, environment interaction is faster, and **model update becomes the main cost — which can also be accelerated with more GPUs** (we used only two A100s for training). Overall, training remains efficient and scalable with appropriate infrastructure.
> >
> > ---
> >
> > **(2) Total Training Time & Steps**
> >
> > We also report **full training durations and optimization steps in [a]**.
> >
> > We summarize below:
> >
> > | Task Category         | Dataset(s)     | Optimization Steps | Total Training Time (min) |
> > | --------------------- | -------------- | ------------------ | ------------------------- |
> > | **Classic IR**        | NFCorpus       | 230                | 228                       |
> > |                       | FEVER          | 600                | 645                       |
> > |                       | HotpotQA       | 620                | 831                       |
> > |                       | SciFact        | 110                | 181                       |
> > |                       | MS-BEIR        | 400                | 2098                      |
> > | **Literature Search** | PubMed         | 600                | 1507                      |
> > |                       | ClinicalTrials | 120                | 352                       |
> > | **Evidence-Seeking**  | NQ             | 500                | 996                       |
> > |                       | TriviaQA       | 360                | 412                       |
> > |                       | SQuAD          | 330                | 401                       |
> > | **SQL Search (3B)**   | BIRD           | 240                | 435                       |
> > |                       | Spider         | 340                | 357                       |
> > | **SQL Search (7B)**   | BIRD           | 300                | 848                       |
> > |                       | Spider         | 260                | 554                       |
> >
> > These show that DeepRetrieval converges quickly (averaged 11.7 hours per task), and runs substantially faster than prior methods like RaFe [4] or LEADS [1] (32x-217x faster than RaFe and 3x-5x faster than LEADS, see [a] for more details).
> >
> >
> >
> > ---
> >
> > **References**
> >
> > [1] Wang, et al. "A foundation model for human-AI collaboration in medical literature mining." *arXiv preprint arXiv:2501.16255* (2025).
> >
> > [2] Xiong et al. "Benchmarking retrieval-augmented generation for medicine." ACL 2024 findings.
> >
> > [3] Karpukhin et al. "Dense passage retrieval for open-domain question answering." EMNLP 2020.
> >
> > [4] Mao et al., "RaFe: Ranking Feedback Improves Query Rewriting for RAG." Findings of EMNLP 2024
> >
> > [a] [(Anonymous) Efficiency & Performance Study](https://docs.google.com/spreadsheets/d/e/2PACX-1vT8QSWPEhfB21q6n3e5j0KEj8RB03wbv4rGVYVEjAxcD5Yn2M1E7wnfiXdPm5_o0feSWqHAH_SR-v8h/pubhtml)
> >
> > [b] [(Anonymous) Runtime Decomposition of DeepRetrieval](https://docs.google.com/spreadsheets/d/e/2PACX-1vT4F38Tw1cAK3ksmG-U6G_afquguxCuBCHJZmC9k6MzUqz0_mdEKnC8elJsp76M7xSemObg8tmuTHBl/pubhtml)

---

> > ### Author Response · Authors · 2025-05-30
> > **[3/3] Addressing W3 – On Dataset Choice of Evidence-Seeking Retrieval: Generalization Beyond NQ/TriviaQA/SQuAD**
> >
> > (continue)
> >
> >
> >
> > > **[W3] Use of outdated QA datasets in evaluation**
> >
> > We agree with your concerns regarding the limitations of NQ, TriviaQA, and SQuAD — especially memorization risks and answer drift over time.
> >
> > To address this, we performed **additional evaluation** on both newer and domain-shifted QA benchmarks. Crucially, we evaluate **QA accuracy** to more directly measure **real-world retrieval utility** (i.e., whether improved search leads to better answers). The benchmarks include:
> >
> > - **Five general-domain QA benchmarks**: *PopQA, 2Wiki, Musique, TriviaQA, HotpotQA*
> > - **Four medical QA benchmarks [2]**: *MedQA-US, MedMCQA, PubMedQA, MMLU-Med*
> >
> > Importantly, we emphasize that the **DeepRetrieval model used here is *exactly the same* model trained on the NQ dataset for the evidence-seeking task** in our original paper. It was not fine-tuned or adapted in any way to these new datasets, demonstrating **zero-shot generalization** purely from reward-guided training on a single dataset.
> >
> > General-domain QA:
> >
> > | Method                    | TriviaQA | PopQA    | HotpotQA | 2wiki    | Musique  |
> > | ------------------------- | -------- | -------- | -------- | -------- | -------- |
> > | Direct Inference          | 76.5     | 35.7     | 35.5     | 28.9     | 8.8      |
> > | RAG-BM25                  | 75.5     | 35.9     | 50.2     | 40.7     | 11.8     |
> > | **DeepRetrieval-BM25-NQ** | **80.2** | **45.5** | **54.5** | **47.1** | **22.2** |
> >
> > Medical-domain QA:
> >
> > | Method                    | MedQA-US | MedMCQA  | PubMedQA | MMLU-Med |
> > | ------------------------- | -------- | -------- | -------- | -------- |
> > | Direct Inference          | 61.7     | 55.8     | 55.6     | 76.4     |
> > | RAG-BM25                  | 61.6     | 57.5     | 52.8     | 77.6     |
> > | **DeepRetrieval-BM25-NQ** | **62.5** | **61.3** | **56.2** | **79.2** |
> >
> > where for RAG-BM25 and DeepRetrieval, we used top 3 retrieved documents as the context fed to generator LLM. We used **Claude-3-Haiku** as the generator LLM, **Wikipedia-2018** [3] as the retrieval corpus, and **LLM-as-a-judge** for answer accuracy, mitigating concerns about surface-form matching or outdated labels.
> >
> > These results demonstrate that:
> >
> > - DeepRetrieval learns **robust, transferable query strategies**;
> > - The method improves retrieval quality even in settings where LLM memorization is unlikely or irrelevant;
> > - Its generalization supports real-world deployment where future questions/data will differ from training.
> >
> > We believe this addresses your concern and strengthens the case for DeepRetrieval as a practical and generalizable approach for real-world RAG applications.
> >
> > ------
> >
> > > **Q1: Which retriever was used in Table 1 for QA tasks?**
> >
> > For the **evidence-seeking QA tasks** (NQ, TriviaQA, SQuAD in Table 1), we use **BM25** as the retriever. This is also clarified in Table 1’s caption.
> >
> > ------
> >
> > > **Q2: Do you concatenate the rewritten query with the original query?**
> >
> > No, we **do not** perform query concatenation. During inference, **only the rewritten query** generated by the model is used for retrieval. This is consistent across all tasks and retrievers.
> >
> > ---
> >
> > **References**
> >
> > [1] Wang, et al. "A foundation model for human-AI collaboration in medical literature mining." *arXiv preprint arXiv:2501.16255* (2025).
> >
> > [2] Xiong et al. "Benchmarking retrieval-augmented generation for medicine." ACL 2024 findings.
> >
> > [3] Karpukhin et al. "Dense passage retrieval for open-domain question answering." EMNLP 2020.
> >
> > [4] Mao et al., "RaFe: Ranking Feedback Improves Query Rewriting for RAG." Findings of EMNLP 2024
> >
> > [a] [(Anonymous) Efficiency & Performance Study](https://docs.google.com/spreadsheets/d/e/2PACX-1vT8QSWPEhfB21q6n3e5j0KEj8RB03wbv4rGVYVEjAxcD5Yn2M1E7wnfiXdPm5_o0feSWqHAH_SR-v8h/pubhtml)
> >
> > [b] [(Anonymous) Runtime Decomposition of DeepRetrieval](https://docs.google.com/spreadsheets/d/e/2PACX-1vT4F38Tw1cAK3ksmG-U6G_afquguxCuBCHJZmC9k6MzUqz0_mdEKnC8elJsp76M7xSemObg8tmuTHBl/pubhtml)
> >
> > ---
> >
> > Please let us know if we can clarify anything further. We’d be happy to provide additional ablations or results as needed!
> >
> > Best regards,
> >
> > *Authors of DeepRetrieval*

---

> > > ### Comment · Reviewer_C5L2 · 2025-06-05
> > > **Raised my scores as my concerns addressed**
> > >
> > > I thank the authors for going to such incredible lengths to address my concerns. I increased my score to 7.

---

> > > > ### Author Response · Authors · 2025-06-05
> > > > **Thank You for Updating Your Score**
> > > >
> > > > **Dear Reviewer C5L2,**
> > > >
> > > > Thank you for your thoughtful feedback and for updating your score. We truly appreciate your recognition and support.
> > > >
> > > > Sincerely,
> > > >
> > > > *Authors of DeepRetrieval*

---

### Official Review · Reviewer_HAPX · 2025-05-12

**Rating:** 8
**Confidence:** 3
**Ethics Flag:** 1

**Summary:**

The paper introduces DeepRetrieval, a reinforcement learning (RL) approach for training Large Language Models (LLMs) to generate queries in various search settings, optimizing for retrieval performance. In contrast to previous approaches, DeepRetrieval uses a purely RL approach, where the models are trained through trial and error, requiring no supervision. Through extensive experimental evaluations, the authors find that DeepRetrieval significantly improves upon many SOTA methods in various search tasks, even surpassing large LLMs such as GPT-4o or Claude-3.5 using models with only 3B parameters.

**Reasons To Accept:**

- The paper has very strong empirical observations, comparing with strong baselines. The authors explored the efficacy of their method in literature search, evidence-seeking retrieval, classic information retrieval, and SQL database search, observing that in almost all the cases, DeepRetrieval managed to achieve remarkable performance, often clearly surpassing the previous SOTA.

- DeepRetrieval is highly adaptable across diverse retrieval settings. The only modification that is need is to modify the reward function. In other words, the same RL framework can be applied to optimize for different retrieval metrics, eliminating the need for task-specific supervision.

- Besides the evaluations on the different tasks, I find the three takeaways of Section 4 insightful. First, the knowledge injection study seems surprising to me. DeepRetrieval learns to adaptively inject more prior knowledge into the queries when beneficial, and less when unnecessary, without being trained towards that objective. Second, the comparison between RL and SFT highlights that RL encourages exploration of the solution space, while SFT encourages pattern memorization of the training data. Finally, the analysis of the reasoning process shows that adding a structured thinking step not only improves performance but also helps the model explore better query strategies more efficiently during training.

**Reasons To Reject:**

- Although the experimental results are extensive, I think that the paper should also include an ablation study that is related to the design of the reward function. For instance, it would be interesting to explore how sensitive DeepRetrieval is for different rewards, or how the performance changes when optimizing for only retrieval metrics versus combining retrieval and format rewards. This could provide deeper insight into which aspects of the reward function are most influential for the observed improvements.

---

> ### Author Response · Authors · 2025-05-31
> **On the Influence of Reward Function Variants in DeepRetrieval**
>
> **Dear Reviewer HAPX,**
>
> Thank you very much for your encouraging review and thoughtful suggestion on reward design. We sincerely appreciate your insights, which prompted us to revisit several reward configurations explored during development. We're glad to report that **DeepRetrieval performs robustly across these variations**, and we will include the results and discussions in the final revision.
>
> ---
>
> > **Literature Search**
>
> We compared optimizing for **Recall@500** versus **Recall@3K**. As shown below, using Recall@500 slightly improves early retrieval, while Recall@3K leads to better overall coverage. **DeepRetrieval performs robustly under both reward choices**, and the trade-off can be tuned based on downstream needs.
>
> |            |                      | **Recall@500** | **Recall@3K**                |
> | ---------- | -------------------- | -------------- | ---------------------------- |
> | **PubMed** | Recall@500 as Reward | **49.1**       | 63.6                         |
> |            | Recall@3K as Reward  | 45.7           | **65.1** (reported in paper) |
>
> > **Classic IR**
>
> We ablated two aspects: the **metric type** (NDCG vs. Recall) and the **cutoff threshold** (K). Results indicate that:
>
> - Varying the cutoff (e.g., NDCG@10 vs. NDCG@1000) has limited effect on both NDCG and Recall.
> - Switching to Recall@10 slightly improves Recall, with only a minor drop in NDCG.
>
> |                 |                     | **NDCG@10**                  | **Recall@10** |
> | --------------- | ------------------- | ---------------------------- | ------------- |
> | **MS-S (BM25)** | NDCG@10 as Reward   | **41.1** (reported in paper) | 57.6          |
> |                 | NDCG@1000 as Reward | 40.2                         | 57.7          |
> |                 | Recall@10 as Reward | 39.7                         | 59.8          |
>
> These results demonstrate that the model’s performance is not highly sensitive to small variations in reward formulation.
>
> > **SQL Search**
>
> We tested augmenting the reward with an additional syntax correctness term (penalizing syntactically invalid SQL). While this encourages well-formed queries, it **slightly hurts final task accuracy**, possibly because syntactic constraints discourage exploratory behavior.
>
> |            |                                   | **Execution Accuracy**       |
> | ---------- | --------------------------------- | ---------------------------- |
> | **Spider** | Execution Accuracy (EA) as Reward | **74.9** (reported in paper) |
> |            | EA + Syntax Correctness as Reward | 72.1                         |
>
> > **Evidence-Seeking Retrieval**
>
> For this setting, our reward is based on whether the retrieved document contains an answer span matching the gold answer. While more advanced rewards (e.g., LLM-as-a-judge) are possible, they are prohibitively expensive given the large number of training queries (e.g., >70k in NQ). Moreover, as you noted, both our knowledge injection analysis (Figure 3) and cross-task generalization results suggest that the current reward signal is already **robust and effective**.
>
> ---
>
> > **Summary of Our Findings:**
>
> - **Goal-aligned reward functions** (e.g., Recall@3K for literature search, Execution Accuracy for SQL) lead to the best performance.
> - **DeepRetrieval is robust** to reward variations, maintaining strong performance even when reward signals are adjusted.
> - **Secondary components** (e.g., syntax correctness for SQL, cutoff variation in IR) have limited impact to the final performance.
>
> ------
>
> We hope this extended analysis addresses your suggestion. We will include the results and discussions in the final revision. Thank you again for your thoughtful feedback and support. Please let us know if you would like any additional results or clarification!
>
> Best regards,
>
>  *Authors of DeepRetrieval*

---

> > ### Comment · Reviewer_HAPX · 2025-05-31
> >
> > Thank you for your response. I think that, indeed, the analysis addresses my suggestion, and thus, I will maintain my positive assessment.

---

> > > ### Author Response · Authors · 2025-06-01
> > > **Thank You for the Positive Assessment**
> > >
> > > **Dear Reviewer HAPX,**
> > >
> > > Thank you for your quick follow-up and for confirming that our analysis addressed your suggestion. We sincerely appreciate your thoughtful feedback and positive assessment. Your support has been encouraging and helpful in improving the paper of DeepRetrieval.
> > >
> > > Sincerely,
> > >
> > > *Authors of DeepRetrieval*

---

> > > > ### Comment · Reviewer_fHLD · 2025-06-07
> > > >
> > > > Thank you for your response, Most of my concerns have been addressed. I am especially impressed by the thorough comparison of the models I referred to. Therefore, I have decided to raise my rating to accept.

---

### Official Review · Reviewer_fHLD · 2025-05-12

**Rating:** 7
**Confidence:** 4
**Ethics Flag:** 1

**Summary:**

This paper presents a RL-based method for training large language models to improve information retrieval through optimized query generation. The main idea is to directly optimize queries based on retrieval performance (e.g., Recall@K, NDCG). Experimental results on four tasks such as literature search, evidence seeking QA, classic IR, and SQL search demonstrate that the proposed method achieves reasonable performance than baselines

**Reasons To Accept:**

- The paper is well-written and easy to follow.

- Extensive experiments are conducted to demonstrate the superiority of the proposed method.

**Reasons To Reject:**

- The paper notes that constructing supervised datasets requires significant computational resources (line 41), yet the proposed RL-based approach and reasoning components may also be resource-intensive. A more thorough analysis or discussion of computational cost is needed.

- While reasoning is incorporated in the proposed method, it is unclear whether the reasoning contributes meaningfully to real-world search tasks. Although execution times for BM25 and dense retrieval are provided, the paper does not report the execution time of the proposed method, making it difficult to assess its practicality.

- The method uses explicit ground truth documents for reward signals, but it lacks comparison with query rewriting models trained on query–ground truth document pairs.

- There is no comparison with existing LLM-based query rewriting methods such as Ma et al. (EMNLP 2023), [1] or models discussed in IR conferences [2], making it difficult to judge the advantage of the proposed approach over current methodologies.
  - [1] Mao et al., RaFe: Ranking Feedback Improves Query Rewriting for RAG, Findings of EMNLP 2024
  - [2] Kostic and Balog, A Surprisingly Simple yet Effective Multi-Query Rewriting Method for Conversational Passage Retrieval, SIGIR 2024

---

> ### Author Response · Authors · 2025-05-29
> **[1/2] Addressing W1 - DeepRetrieval is More Efficient than "Supervised Data Collection" Pipelines**
>
> **Dear Reviewer fHLD,**
>
> We sincerely appreciate your thoughtful review! To address the issues you raised, we have taken the following steps over the past few days:
>
> 1. We conducted a **comprehensive efficiency and performance study** for DeepRetrieval and the SFT baselines, using the same hardware and software setup. Please refer to the raw results in [a].
> 2. We **implemented and ran RaFe (EMNLP 2024 [1]) and CMQR (SIGIR 2024, [2]) since neither provides open-source code**. Our implementation and logs are in [b]; performance comparisons with DeepRetrieval can be found in [a].
>
> Below, we explain how these results directly address your concerns.
>
> ---
>
> > **[W1]** The paper notes that constructing supervised datasets requires significant computational resources (line 41), yet the proposed RL-based approach and reasoning components may also be resource-intensive. A more thorough analysis or discussion of computational cost is needed.
>
> According to our new results [a]:
>
> **Classic IR:**
>
> | Method        |                  | NFCorpus | FEVER   | HotpotQA | SciFact | MS-BEIR  |
> | ------------- | ---------------- | -------- | ------- | -------- | ------- | -------- |
> | RaFe [1]      | Train Time (min) | 6598     | 279730  | 216529   | 2084    | 1281185  |
> | DeepRetrieval | Train Time (min) | **228**  | **645** | **831**  | **181** | **2098** |
>
> We estimate RaFe's training time per batch (64 samples) based on our logs (*rafe_training.log* in [b]):
>
> $\frac{654\text{s} \text{(Rewrite Data Generation)} + 290\text{s} \text{(SFT)} + 6281\text{s} \text{(Feedback Generation)} + 2557\text{s} \text{(DPO)}}{60 \text{s/min}} = 163 \text{min} \text{(per batch)}$
>
> In contrast, DeepRetrieval requires only **0.75 to 5.1 minutes per batch**, compared to RaFe’s **163 minutes per batch**. This efficiency comes from our streamlined training: **RaFe uses four stages** (data generation, SFT, feedback scoring, RL), while **DeepRetrieval uses a single PPO (RL) step**, directly optimizing the reward. Due to RaFe's high cost, we couldn't finish training on all IR datasets.
>
> **Literature Search (Search Engines):**
>
> | Method        |                  | PubMed                                      | ClinicalTrials                              |
> | ------------- | ---------------- | ------------------------------------------- | ------------------------------------------- |
> | LEADS [3]     | Train Time (min) | 853 (Distill from GPT) + 3600 (SFT)  = 4453 | 342 (Distill from GPT) + 1500 (SFT)  = 1842 |
> | DeepRetrieval | Train Time (min) | **1507**                                    | **352**                                     |
>
>
> Thus, we conclude that **DeepRetrieval is significantly more efficient than "supervised data collection + SFT" approaches**.
>
> ---
> **References**
>
> [1] Mao et al., RaFe: Ranking Feedback Improves Query Rewriting for RAG, Findings of EMNLP 2024
>
> [2] Kostic and Balog, A Surprisingly Simple yet Effective Multi-Query Rewriting Method for Conversational Passage Retrieval, SIGIR 2024
>
> [3] Wang, et al. "A foundation model for human-AI collaboration in medical literature mining." *arXiv preprint arXiv:2501.16255* (2025).
>
> [a] [(Anonymous) Efficiency & Performance Study](https://docs.google.com/spreadsheets/d/e/2PACX-1vT8QSWPEhfB21q6n3e5j0KEj8RB03wbv4rGVYVEjAxcD5Yn2M1E7wnfiXdPm5_o0feSWqHAH_SR-v8h/pubhtml)
>
> [b] [(Anonymous) Implementation and Experiment Logs of RaFe [1] and CMQR [2]](https://drive.google.com/drive/folders/1SiwhzlKLX_EAxWe89FPqBlI8HXBtAMyV?usp=sharing)

---

> > ### Author Response · Authors · 2025-05-29
> > **[2/2] Addressing W2–W4: DeepRetrieval’s Reasoning Design, Practical Efficiency, and Empirical Superiority**
> >
> > (continue)
> >
> > > **[W2.1]** While reasoning is incorporated in the proposed method, it is unclear whether the reasoning contributes meaningfully to real-world search tasks.
> >
> > As mentioned in our paper (paragraph from Line 307), we conclude that *reasoning is not essential for solving the task itself, but rather served as an auxiliary tool for exploration* (Line 345-347). **the model tends to indiscriminately expand the query space, leading to degraded performance**, as demonstrated in Figure 12 (page 29) and Figure 4(c). In real-world deployment, we use only the generated query for retrieval, not the reasoning chain.
> >
> >
> >
> > > **[W2.2]** Although execution times for BM25 and dense retrieval are provided, the paper does not report the execution time of the proposed method, making it difficult to assess its practicality.
> >
> > We report the training execution time in our new results [a] and [W1], which shows that our training speed is much faster (**32x-217x faster than RaFe [1] and 3x-5x faster than LEADS [3]**).
> >
> > **For inference**, On a single A6000 GPU, DeepRetrieval takes **0.35 sec/query** (avg. over 1,000 test queries), which is **faster** than CMQR [2] at **0.51 sec/query** on the same setup.
> >
> >
> >
> > > **[W3&W4]** There is no comparison with existing LLM-based query rewriting methods such as Ma et al. (EMNLP 2023), [1] or models discussed in IR conferences [2].
> >
> > Thank you for highlighting these works! As both papers lack open-source code, we reimplemented the methods from their descriptions (see our code and logs in [b]) and report full results in [a]. As described in [W1], RaFe [1] is too resource-intensive to show their results, we summarize the performance of CMQR [2] on Classic IR and Evidence-Seeking Retrieval below.
> >
> > **Classic IR (NDCG@10):**
> >
> > | Method        | NFCorpus | FEVER    | HotpotQA | SciFact  | MS-BEIR  |
> > | ------------- | -------- | -------- | -------- | -------- | -------- |
> > | CMQR [2]      | 28.9     | 43.4     | 54.7     | 55.9     | 45.2     |
> > | DeepRetrieval | **34.0** | **66.4** | **63.1** | **64.6** | **53.1** |
> >
> > **Evidence-Seeking Retrieval (Hits@1, 5, 20):**
> >
> > | Method        | NQ                     | TriviaQA               | SQuAD                  |
> > | ------------- | ---------------------- | ---------------------- | ---------------------- |
> > | CMQR [2]      | (21.7, 43.9, 62.6)     | (41.1, 59.2, 70.6)     | (33.7, 53.8, 67.7)     |
> > | DeepRetrieval | **(35.5, 57.5, 72.7)** | **(58.4, 73.2, 80.6)** | **(38.5, 59.4, 72.9)** |
> >
> >
> > ---
> >
> > **References**
> >
> > [1] Mao et al., RaFe: Ranking Feedback Improves Query Rewriting for RAG, Findings of EMNLP 2024
> >
> > [2] Kostic and Balog, A Surprisingly Simple yet Effective Multi-Query Rewriting Method for Conversational Passage Retrieval, SIGIR 2024
> >
> > [3] Wang, et al. "A foundation model for human-AI collaboration in medical literature mining." *arXiv preprint arXiv:2501.16255* (2025).
> >
> > [a] [(Anonymous) Efficiency & Performance Study](https://docs.google.com/spreadsheets/d/e/2PACX-1vT8QSWPEhfB21q6n3e5j0KEj8RB03wbv4rGVYVEjAxcD5Yn2M1E7wnfiXdPm5_o0feSWqHAH_SR-v8h/pubhtml)
> >
> > [b] [(Anonymous) Implementation and Experiment Logs of RaFe [1] and CMQR [2]](https://drive.google.com/drive/folders/1SiwhzlKLX_EAxWe89FPqBlI8HXBtAMyV?usp=sharing)
> >
> > ---
> >
> > Please let us know if you have any further concerns. We're ready to clarify or provide additional results if needed!
> >
> > Best regards,
> >
> >  *Authors of DeepRetrieval*

---

> > > ### Author Response · Authors · 2025-06-06
> > > **Follow-Up on Rebuttal**
> > >
> > > **Dear Reviewer fHLD,**
> > >
> > > We wanted to kindly check in and see if you had a chance to review our response. We truly appreciate your thoughtful comments and would be happy to run additional experiments or provide clarifications if needed.
> > >
> > > Thank you for your time and feedback!
> > >
> > > Best regards,
> > >
> > > *Authors of DeepRetrieval*

---

> > > > ### Author Response · Authors · 2025-06-07
> > > > **Appreciation for Updated Assessment**
> > > >
> > > > **Dear Reviewer fHLD,**
> > > >
> > > > Thank you for your thoughtful feedback and for raising your score! We're glad our updates addressed your concerns, and we truly appreciate your support in strengthening the paper.
> > > >
> > > > Sincerely,
> > > >
> > > > *Authors of DeepRetrieval*

---

### Decision · Program_Chairs · 2025-07-08

**Decision:**

Accept

**Comment:**

The authors apply PPO for query generation (for IR and SQL) on recent small open LMs, with an extensive study of tradeoffs and ablations.

As reviewer C5L2 says, the authors seem to greatly overstate the "without requiring supervised data for query generation" point in a way that makes it hard to assess in what way the work is novel, even though the extensiveness of the tasks used is interesting.

**Compared to work on neural retrieval:** The vast majority of neural retrieval work (e.g., methods that train *encoders* like the ones the authors cite, Karpukhin et al or Khattab et al, but really just the bulk of the OpenQA and neural retrieval literatures from 2018-2024!) assume nothing but the same ground-truth documents (or gold-answer short string) that the authors here assume for calculating their rewards. Moreover, I suspect that on a numerical comparison head-to-head on these tasks with numbers from literature as old as 2020, the best such encoder methods would win. I don't consider that an argument against this work, but it's an argument against the stated positioning of "learning to retrieve without expensive supervision".

**Compared to work on query generation:** The claims the authors make are less severely affected for query generation, but it is worth noting that some of the papers cited in this work that do query generation (e.g., Hsu et al's "Grounding by trying: LLMs with reinforcement learning-enhanced retrieval"), which the authors cite as requiring "more supervision", also assume nothing but the very same context labels for computing their rewards. They do not assume ground-truth queries.

Thus, some kind of restarting of the precise novel contribution here is necessary. All reviewers are unanimous in recommending acceptance and I will agree with them as a "Maybe accept, if there is room", on the basis that this is the richest work I've seen that applies PPO to query generation in a uniform way across many tasks without having to tune the underlying retrieval mechanism.

Minor: This all challenges the name DeepRetrieval a bit, since it's more like Reinforced Query Generation. It's not deep in the sense of iterative search and it's not about "retrieval" in the sense that the main special characteristic is the ability to use a pre-existing Search Engine without tuning the core retrieval aspects.